# Spatiotemporal contact between peroxisomes and lipid droplets regulates fasting-induced lipolysis via PEX5

Jinuk Kong[1], Yul Ji[1], Yong Geun Jeon[1], Ji Seul Han[1], Kyung Hee Han[1], Jung Hyun Lee[1], Gung Lee[1], Hagoon Jang[1], Sung Sik Choe[1], Myriam Baes[2] & Jae Bum Kim [1]*

Lipid droplets (LDs) are key subcellular organelles for regulating lipid metabolism. Although several subcellular organelles participate in lipid metabolism, it remains elusive whether physical contacts between subcellular organelles and LDs might be involved in lipolysis upon nutritional deprivation. Here, we demonstrate that peroxisomes and peroxisomal protein PEX5 mediate fasting-induced lipolysis by stimulating adipose triglyceride lipase (ATGL) translocation onto LDs. During fasting, physical contacts between peroxisomes and LDs are increased by KIFC3-dependent movement of peroxisomes toward LDs, which facilitates spatial translocations of ATGL onto LDs. In addition, PEX5 could escort ATGL to contact points between peroxisomes and LDs in the presence of fasting cues. Moreover, in adipocyte-specific PEX5-knockout mice, the recruitment of ATGL onto LDs was defective and fasting-induced lipolysis is attenuated. Collectively, these data suggest that physical contacts between peroxisomes and LDs are required for spatiotemporal translocation of ATGL, which is escorted by PEX5 upon fasting, to maintain energy homeostasis.

[1] National Creative Research Initiatives Center for Adipose Tissue Remodeling, Institute of Molecular Biology and Genetics, Department of Biological Sciences, Seoul National University, Seoul, South Korea. [2] Laboratory of Cell Metabolism, Department of Pharmaceutical and Pharmacological Sciences, KU Leuven, Leuven, Belgium. *email: jaebkim@snu.ac.kr

The major role of adipocytes is to store excess energy in the form of lipid droplets (LDs) containing triacylglycerols (TGs)[1,2]. When energy demand increases, catabolic processes start with the initiation of lipolysis; the stored TGs are hydrolyzed into free fatty acids (FFAs) and glycerols. As dysregulation of lipid metabolism is closely linked to pathological conditions including obesity, diabetes, and related metabolic diseases, lipolysis needs to be tightly regulated in response to nutritional status[3]. Lipolysis is primarily mediated by neutral TG hydrolases including adipose triglyceride lipase (ATGL) and hormone-sensitive lipase (HSL)[4–6]. Depending on the nutritional status, lipolysis is tightly regulated by protein kinase A (PKA), which is composed of catalytic subunits and cAMP-binding regulatory subunits[7]. For instance, upon fasting or hormonal stimulation, activated PKA phosphorylates perilipin1 (PLIN1) and HSL. Phosphorylated PLIN1 releases comparative gene identification-58 (CGI-58; Abhd5), a coactivator of ATGL, to stimulate lipolysis upon PKA activation[8,9]. It has been reported that both ATGL and HSL appear to translocate to LDs upon β-adrenergic activation[10,11]. However, the mechanism underlying lipase translocation to LDs remains unclear.

Peroxisomes are important for the degradation of toxic hydrogen peroxide, and they contribute to the maintenance of cellular redox balance[12–14]. In addition, peroxisomes participate in lipid metabolism by oxidizing very long-chain fatty acids, branched-chain fatty acids, and several long-chain fatty acids[15,16] and synthesizing certain lipids, including ether lipids, cholesterol, and bile acid[16,17]. Furthermore, peroxisomes promote fatty acid oxidation to supply energy sources during starvation[18,19]. In this regard, it has been reported that patients with peroxisomal disorders and peroxisome-deficient mice exhibit severe hepatic dysfunctions, including fatty liver, cholestasis, and eventually cirrhosis[20,21].

In eukaryotes, various subcellular-organelles cooperate and communicate with neighboring organelles through physical contacts to coordinate their cellular functions including lipid metabolism. For example, the endoplasmic reticulum contacts with LDs and peroxisomes to promote growths of LDs and peroxisomes[22–26]. Peroxisomes make contact with mitochondria to transfer metabolic products of fatty acid oxidation from peroxisomes to mitochondria[27]. Although it has been reported that peroxisomes appear to physically interact with LDs[28,29], the physiological significance of the contacts between peroxisomes and LDs (PER–LD) in lipid homeostasis is largely unknown. Given that PER–LD is involved in lipid metabolism, it is feasible to speculate that the physical interaction between these two organelles may be implicated in lipid trafficking and energy homeostasis.

The physiological roles of peroxisomes and peroxisomal proteins in response to energy status have been investigated in *Caenorhabditis elegans*, mammalian adipocytes, and adipocyte-specific PEX5-knockout (PEX5 AKO) mice. In this study, we found that the physical contact between PER–LD is crucial in mediating fasting-induced lipolysis. Through RNA interference (RNAi) screening in *C. elegans*, we elucidated that the peroxisomal protein PEX5 mediates fasting-induced lipolysis by escorting ATGL onto LDs, which requires PKA signaling. Collectively, our data suggest that the inter-organelle crosstalk between PER–LD is crucial in the regulation of energy homeostasis in response to energy status.

## Results

### Fasting stimulates the interaction between PER–LD. Although it has been shown that peroxisomes are able to interact with LDs[28], it is unclear which physiological cues induce such physical

contacts. Given that fasting activates catabolic metabolism, including lipolysis and fatty acid oxidation[18,30], we hypothesized that fasting might affect the interaction between PER–LD. To address this hypothesis, we examined the subcellular localization of PER–LD in *C. elegans* in response to nutritional status. Consistent with previous reports[31,32], LDs in the anterior intestine were decreased by fasting (Supplementary Fig. 2a, b). Fasting rapidly stimulated the colocalization of red fluorescence protein (RFP)-tagged peroxisome targeting sequence (PTS), a peroxisome marker[33,34], onto LDs in the intestines of live worms assessed by coherent anti-stokes Raman scattering (CARS) microscopy, without significant changes in peroxisome size (Fig. 1a–c, and Supplementary Fig. 1c). To confirm this observation in mammals, immunohistochemical analysis was conducted with mouse epididymal white adipose tissue (eWAT). As shown in Fig. 1d, peroxisomal membrane protein (PMP) 70, another peroxisome marker, was abundantly detected on LDs upon fasting. To gain further insights into the interaction between PER–LD, differentiated adipocytes were treated with isoproterenol (ISO), a β-adrenergic receptor agonist, to mimic fasting stimuli. In the presence of ISO, the colocalization of PER–LD in adipocytes was greatly enhanced, with little changes in peroxisome size (Fig. 1e and Supplementary Fig. 1d). Consistent herewith, three-dimensional super-resolution microscopy (3D-SIM) revealed that peroxisomes abundantly surrounded the surfaces of LDs in ISO-treated adipocytes (Fig. 1f). Although the total amount of PMP70 was not increased in ISO-treated adipocytes (Fig. 1g), the ratio of colocalization of PMP70 and PLIN1 was elevated by ISO (Fig. 1h). In parallel, the localization of peroxisomal catalase was increased at the surface of LDs upon ISO treatment (Supplementary Fig. 1e). Next, to determine whether peroxisomes would indeed translocate onto LDs upon fasting, we traced the movement of peroxisomes using live imaging. In adipocytes, forskolin (FSK), a pharmacological activator of PKA, promoted the translocation of mCherry-PTS onto LDs (Supplementary Fig. 1f, Supplementary Videos 1, 2, and 3). In accordance herewith, the levels of PMP70 protein were increased in the LD fraction of ISO-treated adipocytes (Fig. 1i). However, unlike peroxisomes, mitochondria did not quickly move toward LDs upon ISO (Supplementary Fig. 1g). These data suggest that fasting would stimulate the physical interaction between peroxisomes and LDs, probably through peroxisome migration.

### PER–LD interaction is crucial for fasting-induced lipolysis. The findings that fasting signals potentiated the movement of peroxisomes to LD led us to investigate whether peroxisomes might move onto LDs via the cytoskeleton and motor protein. As microtubules regulate peroxisome movement[35], the effect of nocodazole, an inhibitor of microtubule polymerization, on peroxisome movement was investigated in adipocytes. As shown in Fig. 2a, b, nocodazole inhibited the translocation of peroxisomes onto LDs upon treatment with ISO. However, the total contents of peroxisomal proteins, such as PMP70, were not altered by nocodazole (Supplementary Fig. 2a). It is of interest to note that the decreased interaction between PER–LD in the presence of nocodazole markedly attenuated lipolytic activity in ISO-treated adipocytes (Fig. 2c). Given that KIFC3, which belongs to the group of C-terminal kinesins, is involved in peroxisome movement[36], we decided to test whether suppression of KIFC3 might affect microtubule-dependent peroxisome movement to LDs. When KIFC3 were suppressed via siRNA, the colocalization of PER–LD in ISO-treated adipocytes was markedly decreased (Fig. 2d, e), without significant changes in peroxisomal PMP70 content (Supplementary Fig. 2b). Similar to the effect of nocodazole treatment, suppression of KIFC3 via siRNA diminished

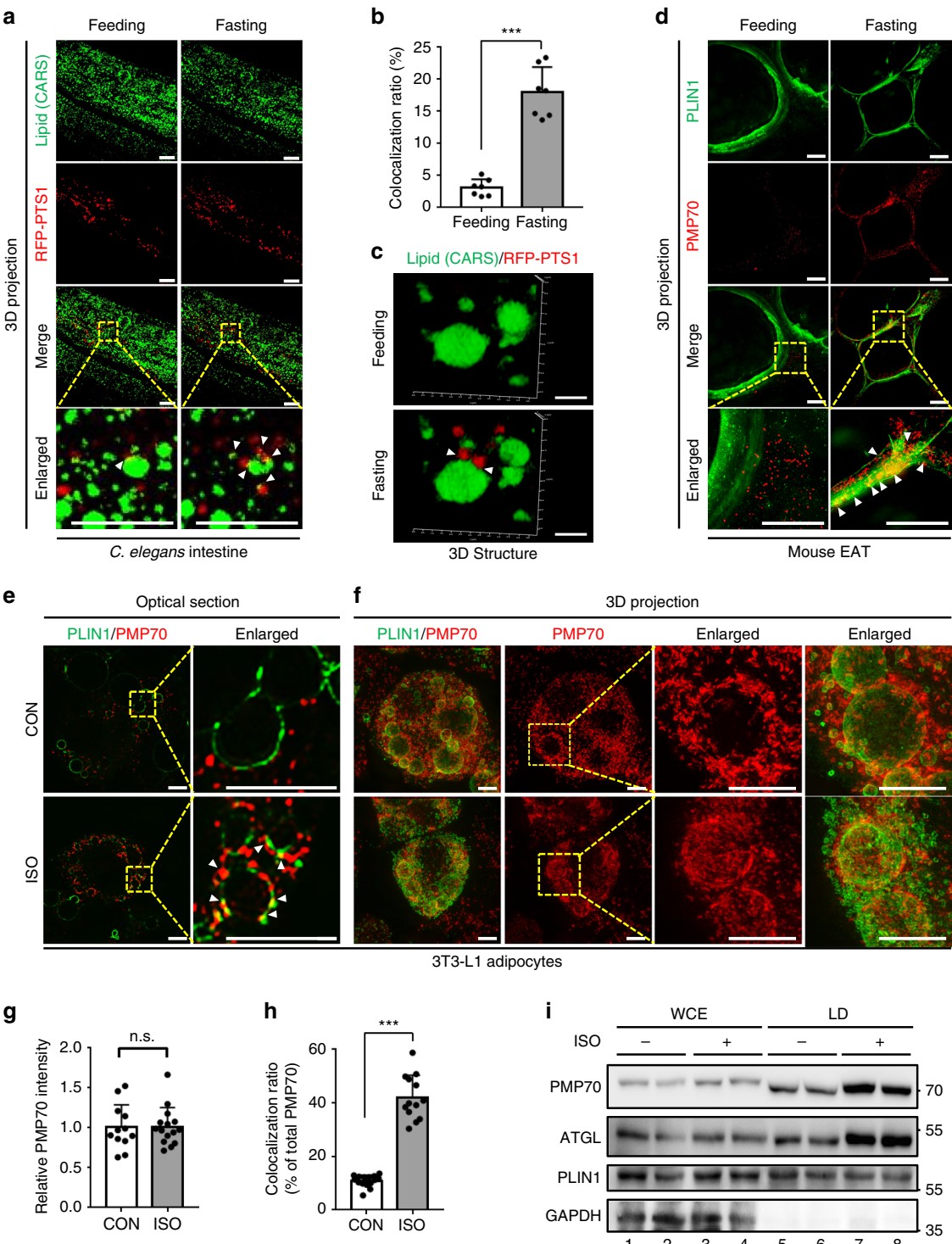

**Fig. 1 Fasting stimuli promote the interaction between PER–LD. a** Representative CARS live images of peroxisome–LD contacts (arrowhead) during fasting (1 h) in young adult worms expressing RFP::PTS1 (peroxisome marker). **b** Quantification of peroxisome–LD colocalization calculated using Leica software (LAS X). $n = 7$ worms. **c** Representative 3D structural images of peroxisome–LD contact (arrowhead) during fasting in *C. elegans*. Scale bars, 5 µm. **d** 3D-SIM images of eWAT immunostained with PLIN1 (LD marker, green) and PMP70 (peroxisome marker, red) under feeding and fasting (12 h) conditions. **e** Representative optical section images of peroxisome–LD contacts (arrowhead) in differentiated 3T3-L1 adipocytes stained for endogenous PMP70 (red) and PLIN1 (green) and treated with ISO (1 µM) for 1 h. **f** 3D projection of SIM images in differentiated 3T3-L1 adipocytes immunostained for PMP70 (Red) and PLIN1 (green) and treated with ISO (1 µM) for 1 h. **g** Quantification of PMP70 staining intensity in adipocytes treated with or without ISO (1 µM) for 1 h. $n = 12$ cells for CON; $n = 15$ cells treated with ISO. **h** Quantification of peroxisome–LD colocalization calculated using imageJ software. $n = 15$ cells for each group. **i** Western blot of whole cell extracts (WCE) or LD fractionation for PMP70 (peroxisome) and LD-associated proteins. 30 µg of protein from WCE; 20 µg of protein from LD fraction. CON control; ISO isoproterenol. All scale bars, 10 µm except for **c**. Arrowhead: contact between PER-LD. Data are represented as the mean ± SD; *$P < 0.05$, ***$P < 0.001$ (unpaired two-tailed Student's *t*-test). n.s., not statistically significant.

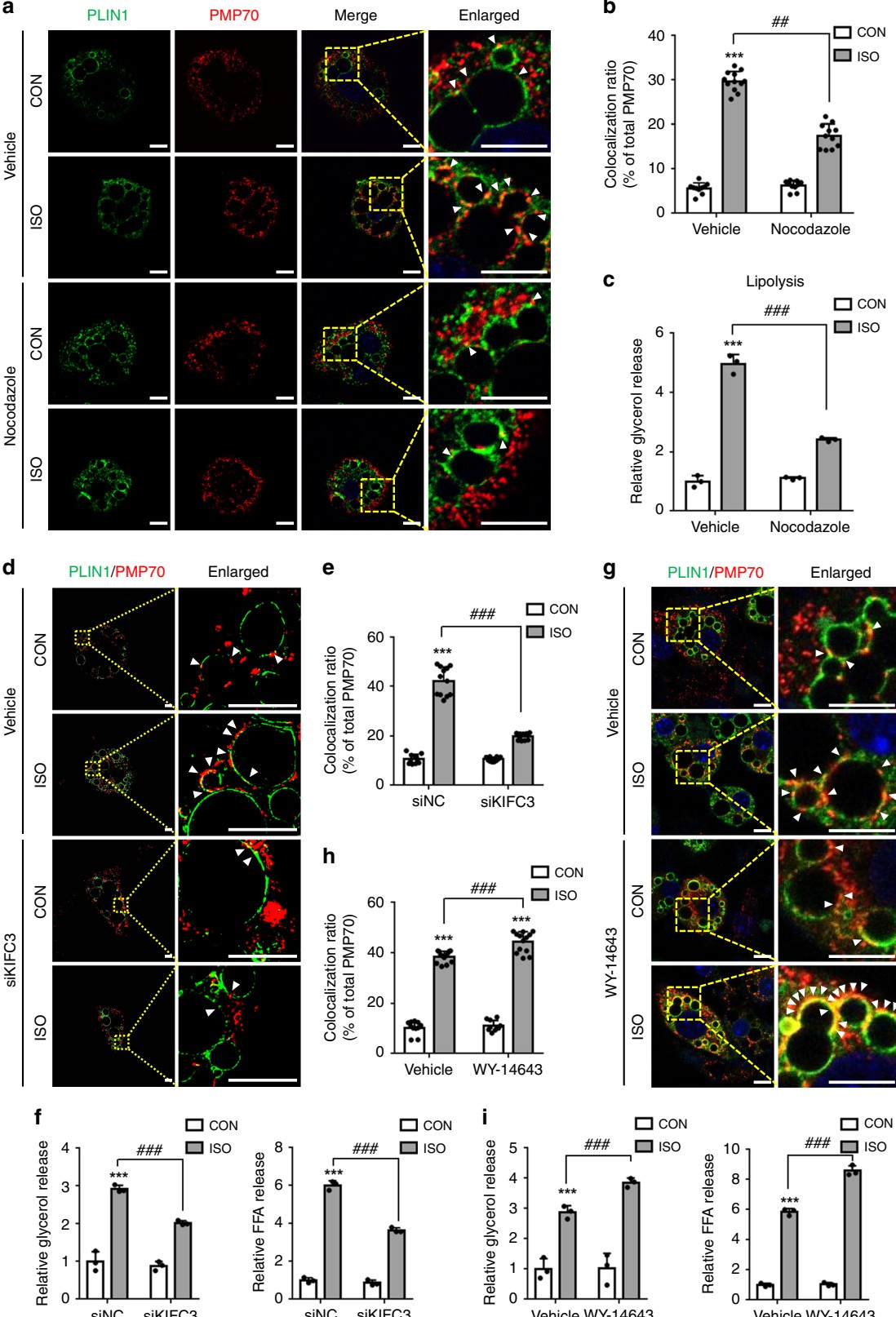

lipolytic activity in ISO-treated adipocytes (Fig. 2f and Supplementary Fig. 2c), implying that the translocation of peroxisomes onto LDs would be important for fasting-induced lipolysis. However, suppression of KIF5B, one of kinesin family, had no effect on peroxisome movement and ISO-stimulated lipolysis (Supplementary Fig. 2d, e and f). Then, to test the idea that the

enhanced interaction between PER–LD could potentiate fasting-induced lipolysis, differentiated adipocytes were treated with WY-14643 (WY), an agonist of peroxisome proliferator-activated receptor α (PPARα). As shown in Fig. 2g, Supplementary Fig. 2g and h, WY augmented the amounts of peroxisomes stained with PMP70 and elevated mRNA levels of PPARα target genes. More

**Fig. 2 Peroxisome–LD contacts are crucial for fasting-induced lipolysis. a**, **b** Representative confocal images and quantification of peroxisome–LD contacts (arrowhead) immunostained with PLIN1 (green) and PMP70 (red) in differentiated adipocytes. Cells were treated with or without nocodazole (0.05 μg ml$^{-1}$) under CON or ISO treatment. $n = 10$ cells for vehicle group; $n = 12$ cells treated with ISO; $n = 10$ cells treated with nocodazole; $n = 11$ cells treated with nocodazole and ISO for quantification of colocalization **b** using imageJ software. **c** Relative glycerol release from adipocytes in media. Adipocytes were treated with or without nocodazole (0.05 μg ml$^{-1}$) under CON or ISO treatment. $n = 3$ for each group. **d**, **e** SIM images and quantitative data of peroxisome–LD contacts (arrowhead) stained with PLIN1 (green) and PMP70 (red) in adipocytes transfected with siNC or siKIFC3 for 48 h. Scale bars, 5 μm. $n = 10$ cells for siNC group; $n = 12$ cells treated with ISO; $n = 10$ cells for siKIFC3 group; $n = 11$ cells for siKIFC3 treated with ISO for quantification of colocalization **e** using imageJ. **f** Relative release of glycerol and FFA from adipocytes transfected with siNC or siKIFC3 for 48 h. $n = 3$ for each group. **g**, **h** Confocal images and quantitative data of peroxisome–LD contacts (arrowhead) immunostained with PLIN1 (green) and PMP70 (red) in adipocytes. Cells were treated with or without WY-14643 (10 μM) for 48 h. $n = 11$ cells for vehicle group; $n = 15$ cells treated with ISO; $n = 10$ cells treated with WY-14643; $n = 13$ cells treated with WY-14643 and ISO for quantification of colocalization **h** using imageJ. **i** Relative release of glycerol and FFA from adipocytes treated with or without WY-14643 (10 μM) for 48 h. $n = 3$ for each group. CON control, ISO isoproterenol. Cells were treated with ISO (1 μM) for 1 h. Arrowhead: contact between PER–LD. Data represent the mean ± SD; $^*P < 0.05$, $^{**}P < 0.01$, $^{***}P < 0.001$ vs. vehicle-CON or siNC-CON, $^{##}P < 0.01$, and $^{###}P < 0.001$ in two-way ANOVA followed by Turkey's post-hoc test. All scale bars, 10 μm except for **d**.

importantly, WY increased the colocalization of PER–LD and enhanced *Atgl* mRNA by ISO (Fig. 2g, h, and Supplementary Fig. 2h). In addition, even though basal lipolytic activity was not altered by WY, ISO-stimulated lipolysis was further elevated by WY (Fig. 2i). These data imply that the physical interaction between PER–LD would be crucial for provoking fasting-induced lipolysis.

**Upon fasting, ATGL translocates to PER–LD contact points**. As peroxisomes moved toward LDs and their interaction promoted fasting-induced lipolysis, we postulated that certain lipases might confer stimulated lipolysis through peroxisome movement. Previously, we have demonstrated that ATGL-1, the ortholog of mammalian ATGL, is a key lipase for fasting-induced lipolysis in *C. elegans*[31]. To test whether fasting-induced peroxisome movement might influence ATGL-1 localization, the subcellular localization of ATGL-1 and peroxisomes in worms was monitored by real-time imaging. Colocalization of ATGL-1 and peroxisomes was promoted during fasting (Fig. 3a, b). In adipocytes, the colocalization of PER–LD was also augmented by ISO (Fig. 3c, d), implying that ATGL might interact with peroxisomes in the presence of fasting stimuli. To determine whether ATGL might be translocated to membrane or matrix of peroxisome, we performed a protease protection assay. As shown in Fig. 3e, the levels of ATGL protein were elevated in the peroxisome fraction of ISO-treated adipocytes. Similar to PMP70, but not peroxisomal matrix protein Catalase, ATGL protein was degraded by proteinase K in both the absence and presence of Triton X-100. Furthermore, when we examined the subcellular location of peroxisomal Catalase and ATGL through super resolution microscopy (SIM), Catalase was detected inside peroxisomes while ATGL was observed at the surface of peroxisomes nearby LDs (Supplementary Fig. 3a, b). Also, we found that the inhibition of peroxisome movement by either nocodazole or KIFC3 siRNA prevented ATGL translocation onto LDs in the presence of ISO (Fig. 3f–i). Together, these data propose that the interaction between PER–LD could mediate the recruitment of ATGL onto LDs upon fasting stimuli.

**PRX5 is required for fasting-induced lipolysis in *C. elegans***. It has been well established that fasting decreases LDs in the anterior intestine of *C. elegans*[31,32,37,38]. We used a reverse genetics approaches to investigate which peroxisomal proteins might be involved in fasting-induced lipolysis. After suppression of specific peroxisomal genes via RNAi, the amounts of LDs in fed and fasted worms were assessed by Oil Red O (ORO) staining. Like in adult wild type (WT) worms, the intestinal ORO staining intensity was reduced upon fasting in most RNAi-treated worms

(Fig. 4a). However, *prx-5* suppression via RNAi significantly attenuated LD hydrolysis upon fasting (Fig. 4a–c). We next examined whether PRX-5 might be associated with ATGL-1-dependent lipolysis. To unveil the genetic interaction between the *prx-5* and *atgl-1* genes, *prx-5* was suppressed via RNAi in ATGL-1 overexpressing worms. While ATGL-1 overexpression decreased intestinal LD in the basal state (Fig. 4d)[31], *prx-5* suppression reversed this effect (Fig. 4d, e). To investigate whether PEX5, the mammalian ortholog of *C. elegans* PRX-5, might be associated with lipolysis in fat tissue, we analyzed the correlations between the expression of *ATGL* and *PEX* genes in human adipose tissues from Genotype-Tissue Expression (GTEx)[39]. As shown in Fig. 4f–h, the level of human *PEX5* mRNA was tightly correlated with that of *ATGL* mRNA in human adipose tissues, similar to our findings in worms. Together, these data propose that the peroxisomal cargo receptor PRX-5/PEX5, together with ATGL, might contribute to mediating fasting-induced lipolysis.

**PEX5 escorts ATGL to LDs for fasting-induced lipolysis**. As a peroxisomal protein, PRX-5 acts as a cargo receptor that recognizes PTSs in target proteins to translocate them to peroxisomes[40,41]. Because peroxisomal biology including the process of protein import may not be the same for mammals and *C. elegans*, the roles of mammalian PEX5 might be different from that of PRX-5 of *C. elegans* in fasting-induced lipolysis. Thus, we decided to investigate the function of mammalian PEX5 in fasting-induced lipolysis further. To this end, we first examined lipolytic activities in adipocytes with or without PEX5 suppression. As shown in Fig. 5a and Supplementary Fig. 4a, PEX5 suppression in adipocytes downregulated ISO-stimulated lipolysis, whereas basal lipolysis was not affected. In the presence of WY, the further increase in ISO-stimulated lipolysis was attenuated by PEX5 suppression (Fig. 5b). Given that PEX5 imports target proteins related to fatty acid oxidation into the peroxisomal matrix[42], we examined whether peroxisomal fatty acid oxidation might also influence PEX5-mediated lipolysis. However, suppression of ACOX1, a key factor for peroxisomal fatty acid oxidation[43], had no effect on fasting-induced lipolysis nor basal lipolysis in adipocytes (Fig. 5c and Supplementary Fig. 4b).

To gain further insights into the roles of mammalian PEX5 in stimulated lipolysis, the subcellular localization of ATGL was examined in adipocytes with or without PEX5 suppression. PEX5 suppression markedly downregulated ISO-induced ATGL translocation onto LDs (Fig. 5d, e). When we analyzed fluorescence intensity line profiles, ATGL and PLIN1 exhibited similar line profiles at LD surfaces in ISO-treated adipocytes (Fig. 5f, Supplementary Fig. 4c). In contrast, PEX5 suppression resulted in distinct ATGL and PLIN1 line profiles. These data suggest that PEX5 would be required for recruiting ATGL onto LDs for stimulated lipolysis. In accordance herewith, ATGL and

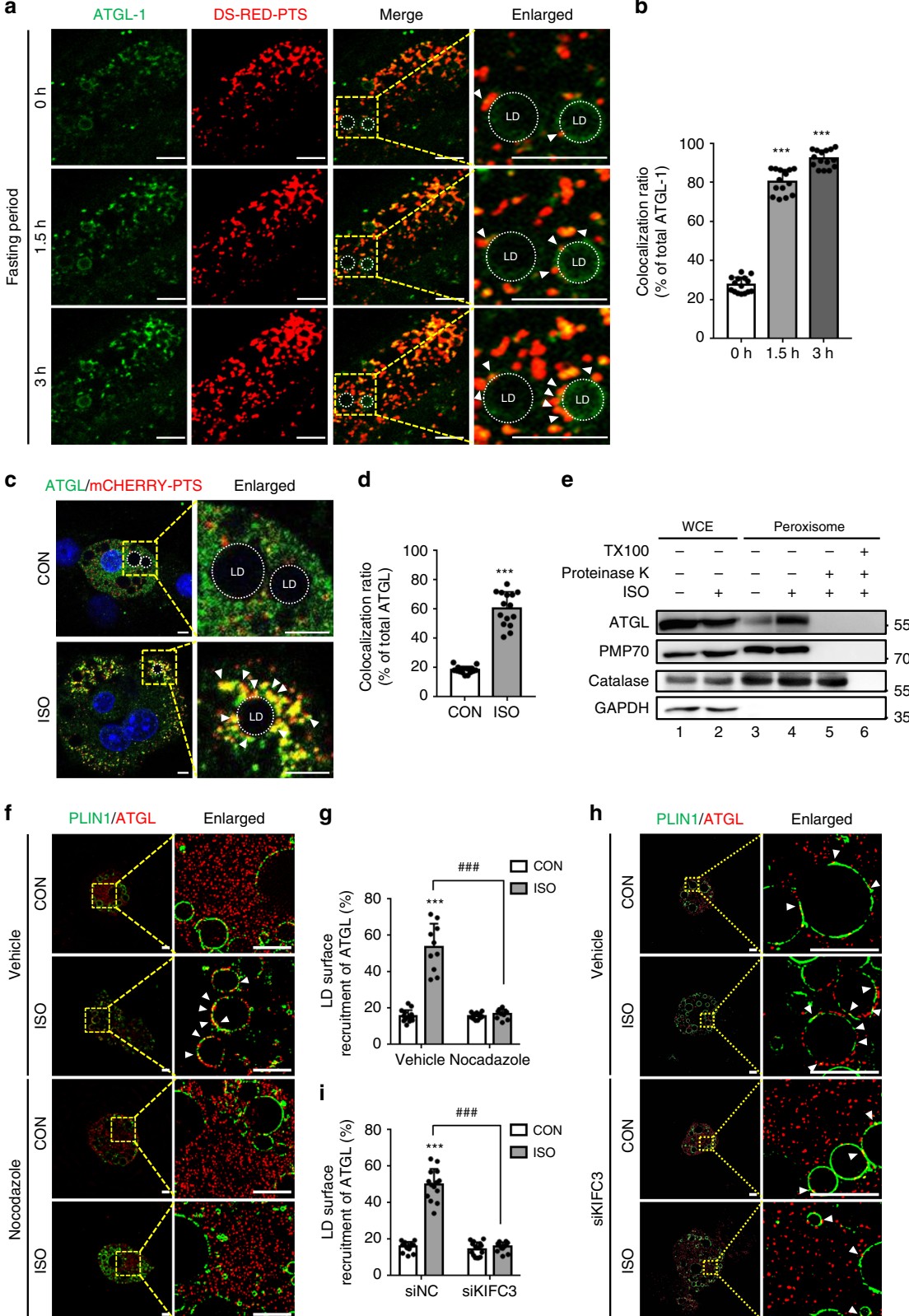

PEX5 proteins were colocalized at the contact points between PER–LD (Supplementary Fig. 4d). On the other hand, ISO-induced interaction between PER–LD was not affected by suppression of PEX5 or ATGL (Supplementary Fig. 4e, f), implying that PEX5 nor ATGL would not be involved in the formation of contact between PER–LD. However, it is likely that PEX5 would be responsible for ATGL translocation onto the contact points between PER–LD. Next, the subcellular distribution of ATGL and PLIN1 proteins was biochemically investigated in adipocytes treated with or without PEX5 siRNA. In accordance with the above data, suppression of PEX5 diminished ATGL translocation into the LD fraction during

**Fig. 3 ATGL translocates to contact sites between PER-LD upon fasting. a** Representative time-lapse confocal images for colocalization of ATGL and peroxisome at LD surfaces (arrowhead) during fasting (3 h) in young adult worms expressing DsRED::PTS1 and ATGL-1::GFP. $n = 15$ for each group. Scale bars, 10 μm. **b** Percentage of ATGL protein colocalizing with peroxisomes calculated using imageJ software. $n = 15$ for each group. **c** Confocal images using adipocytes transfected with mCHERRY::PTS and stained with endogenous ATGL (green), and treated with ISO (1 μM) for 1 h. Arrowhead: colocalzation of ATGL and peroxisome at LD surfaces. DAPI, blue. Scale bars, 10 μm. **d** Quantification analysis of ATGL colocalized to peroxisomes calculated using imageJ software. $n = 15$ for each group. **e** Protease protection assays performed in the absence or presence of Triton X-100. Proteinase K was treated for 30 min in ice. 30 μg of protein from WCE; 20 μg of protein from peroxisome fraction. **f, g** Representative SIM images and quantification analysis recruited ATGL onto LD surfaces in adipocytes immunostained with endogenous PLIN1 (green) and ATGL (red). Cells were treated with nocodazole (0.05 μg ml⁻¹) under CON or ISO (1 μM) treatment. Scale bars, 5 μm. $n = 15$ cells for vehicle group; $n = 10$ cells treated with ISO; $n = 11$ cells treated with nocodazole; $n = 10$ cells treated with nocodazole and ISO. Arrowhead: ATGL recruited to LD surfaces in **f. h, i** SIM images and quantification analysis of recruited ATGL onto LD surfaces in adipocytes immunostained with endogenous PLIN1 (green) and ATGL (red). Cells were **c** or transfected with siNC or siKIFC3 for 48 h **d** under CON or ISO (1 μM) treatment. Arrowhead: ATGL recruited to LD surfaces in **h.** $n = 13$ cells for siNC group; $n = 14$ cells treated with ISO; $n = 15$ cells for siKIFC3 group; $n = 14$ cells for siKIFC3 treated with ISO. Scale bars, 5 μm. ATGL recruitment was quantified using imageJ. CON control; ISO isoproterenol. Data represent the mean ± SD; \*\*\*$P < 0.001$ vs. vehicle-CON or siNC-CON or fasting 0 h, and ###$P < 0.001$ group in two-way ANOVA followed by Turkey's post-hoc test.

ISO-stimulated lipolysis (Fig. 5g, h). Taken together, these data suggest that mammalian PEX5 would play a crucial role in fasting-induced lipolysis by recruiting ATGL onto LDs.

**PEX5 mediates stimulated lipolysis through interacting with ATGL.** To study the subcellular localization of ATGL and PEX5 upon catabolic signals, differentiated adipocytes were treated with or without FSK. Consistent with the above findings, ATGL proteins were abundantly present in the cytosol under the basal conditions, whereas ATGL largely localized to the LD surfaces upon treatment with FSK (Fig. 6a). Similarly, the majority of PEX5 protein was found on LD surfaces upon treatment with FSK. Co-immunoprecipitation assays revealed the biochemical interaction between ATGL and PEX5 (Fig. 6b). Since ATGL does not have C-terminal PTS1 motif, it seems that PEX5 would recognize ATGL independent of canonical PTS1. As the colocalization of ATGL and PEX5 appeared to be increased by FSK (Fig. 6a), we next investigated the degree of biochemical interaction between ATGL and PEX5. Consistent with the above data, the interaction between ATGL and PEX5 was further elevated by FSK (Fig. 6c). Similarly, in situ proximity ligation assay (PLA) affirmed the interaction between ATGL and PEX5 (Fig. 6d). Further, FSK treatment potentiated the interaction in the cytoplasm of HEK293 cells (Fig. 6d). Next, to test whether increased interaction between PEX5 and ATGL would affect stimulated lipolysis, the level of glycerol release was examined with or without FSK. As shown in Fig. 6e, stimulated lipolysis with FSK was substantially augmented by overexpression of ATGL and PEX5, implying that the interaction of ATGL and PEX5 would be important to mediate stimulated lipolysis. Then, we asked whether PKA activation upon catabolic signals might modulate this interaction. As shown in Fig. 6f, FSK increased PEX5 phosphorylation and enhanced the interaction with ATGL. On the contrary, FSK-induced interaction between ATGL and PEX5 was mitigated by phosphatase treatment. These data imply that PEX5 would recruit ATGL to modulate fasting-induced lipolysis, which seems to be promoted by PKA activation.

**Fasting-induced lipolysis is impaired in PEX5 AKO mice.** To elucidate the physiological roles of mammalian PEX5 upon nutritional deprivation, we decided to generate adipocyte-specific PEX5 knockout (PEX5 AKO) mice by crossing adiponectin-cre mice with PEX5-loxP mice in C57BL/6 background. As shown in Fig. 7a, body weight was not altered by PEX5 ablation. However, compared to WT mice, PEX5 AKO mice showed less body weight reduction upon fasting (Fig. 7b). Likewise, the masses of eWAT and inguinal white adipose tissue (iWAT) were less decreased in

fasted PEX5 AKO mice than in fasted WT mice (Fig. 7c). Next, when we examined adipocyte size upon nutritional state, PEX5 ablation greatly attenuated the fasting-induced reduction in adipocyte size (Fig. 7d, e). Then, to examine whether fasting-induced lipolysis might be altered in PEX5 AKO mice, ATGL recruitment onto LDs was compared in adipose tissues of WT and PEX5 AKO mice upon fasting. As shown in Fig. 7f, g, ATGL largely localized on LD surfaces in adipocytes of fasted WT mice. In contrast, PEX5-ablated adipocytes were impaired to translocate ATGL onto LDs in response to fasting. In accordance herewith, serum levels of FFA and glycerol were lower in fasted PEX5 AKO mice than in fasted WT mice (Fig. 7h, i). Furthermore, to determine whether the reduced levels of FFA and glycerol in fasted PEX5 AKO mice might be resulted from decreased lipolytic activity, the effect of ATGL inhibition on PEX5-mediated lipolysis was investigated in adipose tissues cultured ex vivo with or without ISO. As indicated in Fig. 7j, ISO-stimulated lipolytic activity was significantly attenuated in PEX5 AKO. On the contrary, PEX5 ablated eWAT did not show inhibitory effects on ISO-stimulated lipolysis when enzymatic activity of ATGL was blocked by pharmacological inhibitor Atglistatin (Supplementary Fig. 5a, b). While cytosolic ATGL of WT adipocytes was translocated onto LD surfaces upon ISO, this translocation was blocked by PEX5 deletion (Fig. 7k, l), implying that adipocyte PEX5 would be crucial for ATGL translocation onto LDs upon fasting stimuli. Nevertheless, the interaction between peroxisomes and LDs upon fasting was not affected by PEX5 ablation (Supplementary Fig. 5c). Also, PEX5 ablation in adipocytes did not alter liver weight and histology (Supplementary Fig. 5d, e). Collectively, these data indicate that adipocyte PEX5 could escort ATGL onto LDs to provoke fasting-induced lipolysis.

**PEX5 regulates lipolysis in a CGI-58-independent manner.** It has been reported that CGI-58 plays a key role in regulating the hydrolase activity of ATGL[9]. To test whether CGI-58 might be involved in PEX5-mediated lipolysis, the levels of stimulated lipolysis in adipocytes were examined with or without suppression of ATGL, PEX5 and/or CGI-58. Similar knockdown efficiency was confirmed (Supplementary Fig. 6d–h). Knockdown of each single gene decreased stimulated lipolysis to a certain degree (Supplementary Fig. 6a, lanes 3, 4, and 6). Among these, ATGL suppression potently blocked stimulated lipolysis (lanes 6, 7, 8, and 9), and co-suppression of PEX5 and ATGL showed no further inhibitory effect on lipolytic activity (lane 7), implying that PEX5 and ATGL might act in the same pathway. The effect of co-suppression of CGI-58 and ATGL was similar to that of PEX5 and ATGL knockdown (lane 8). On the other hand, PEX5 and CGI-58 co-suppression revealed slight but

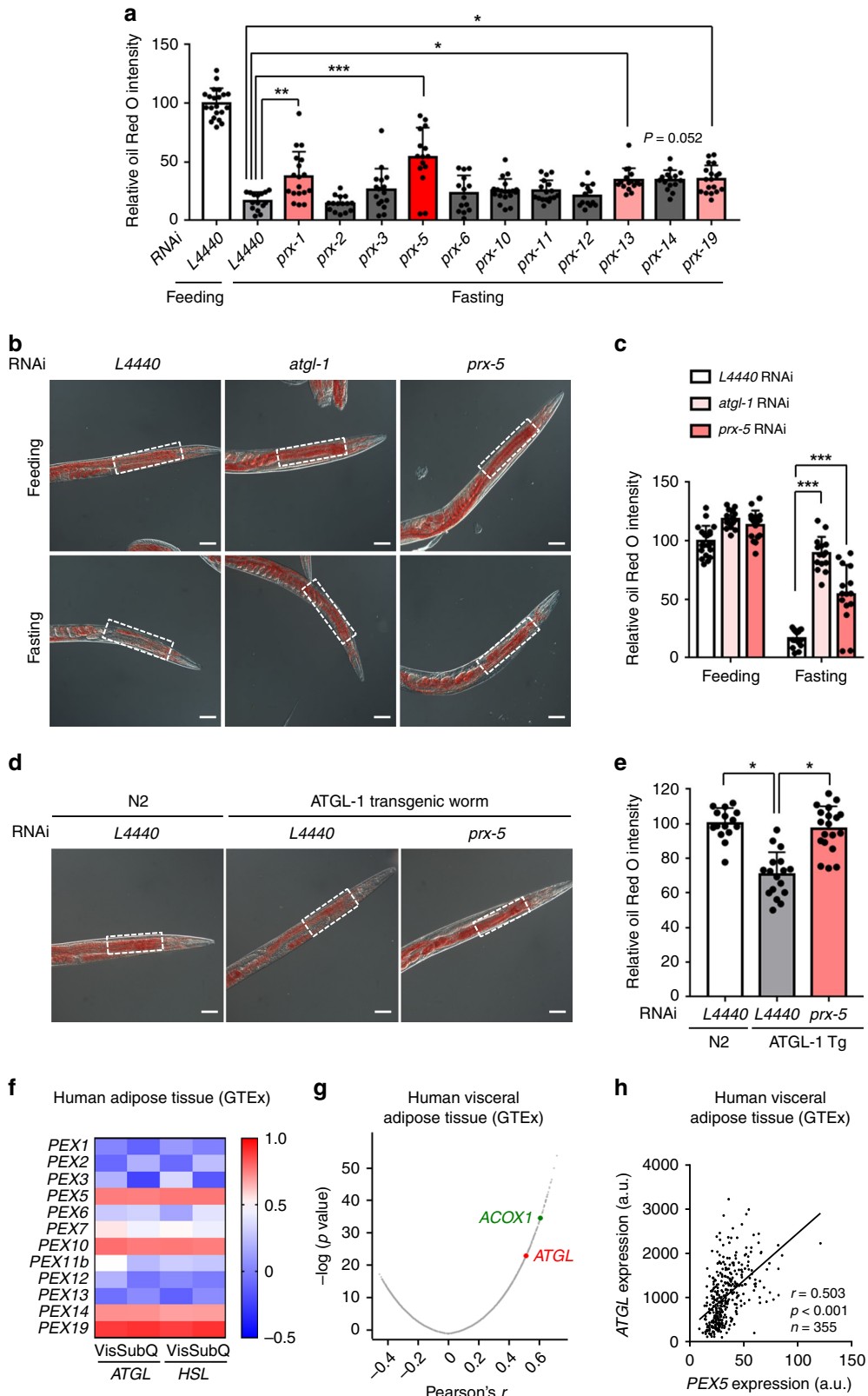

statistically additional effects on stimulated lipolysis (lane 5), implying that PEX5 and CGI-58 might regulate ATGL-mediated lipolysis probably through different pathways (lanes 3, 4, and 5). Moreover, adipocyte CGI-58 dissociated from the LD surfaces and dispersed into the cytoplasm upon treatment with ISO (Supplementary Fig. 6b), which was quite different from the

pattern of PEX5 (Fig. 5). To examine whether CGI-58 would influence the translocation of ATGL onto LDs upon fasting signal, the subcellular localization of ATGL was investigated in CGI-58-suppressed adipocytes. As shown in Supplementary Fig. 6c, CGI-58 suppression did not affect ATGL translocation onto LDs upon ISO. These data imply that PEX5 and CGI-58

**Fig. 4 PRX-5 is required for fasting-induced lipolysis in *C. elegans*. a** RNAi screening of peroxisomal genes involved in fasting-induced lipolysis-based Oil Red O (ORO) staining in anterior intestine of *C. elegans*. ORO staining intensities in young adult RNAi-treated worms under feeding and 8-h fasting conditions were quantified and classified according to the relative fold increase compared to the *L4440* control group. *n* = 21 for feeding *L4440*; *n* = 15 for fasting *L4440*; *n* = 18 for *prx-1*; *n* = 15 for *prx-2*; *n* = 16 for *prx-3*; *n* = 15 for *prx-5*; *n* = 14 for *prx-6*; *n* = 19 for *prx-10*; *n* = 17 for *prx-11*; *n* = 13 for *prx-12*; *n* = 16 for *prx-13*; *n* = 15 for *prx-14*; *n* = 18 for *prx-19*. **b, c** Representative images and quantification of ORO staining in anterior intestine of *C. elegans* with RNAi of *atgl-1* and *prx-5* in young adult worms under feeding and fasting (8 h). *n* = 15–20 for quantification. *n* = 21 for feeding *L4440*; *n* = 16 for feeding *atgl-1*; *n* = 16 for *prx-5*; *n* = 15 for fasting *L4440*; *n* = 15 for fasting *atgl-1*; *n* = 15 for fasting *prx-5*. **d, e** Representative images and quantification of ORO staining in anterior intestine from *prx-5* RNAi-treated WT worms (N2) and *atgl-1* transgenic worms (ATGL-1 Tg, *hj67; Is[atgl-1p::atgl-1::GFP]*). *n* = 15 for *L4440* in N2 worms; *n* = 17 for *L4440* in ATGL-1 Tg; *n* = 19 for *prx-5* in ATGL-1 Tg. **f** Heatmap analysis of Pearson's coefficients (*r*) between lipolytic genes (*ATGL* and *HSL*) and *PEX* genes in human adipose tissue based on data from GTEx. Vis visceral; SubQ subcutaneous. **g** Plots of correlation between *PEX5* and all detectable genes in human visceral adipose tissue based on data from GTEx. *ACOX1* (green), positive control for correlation of *PEX5*. **h** Correlations between expression of *PEX5* and *ATGL* in human visceral adipose tissue based on data from GTEx. All scale bars, 50 μm. Data represent the mean ± SD; *$^*$P < 0.05, $^{**}$P < 0.01, $^{***}$P < 0.001 in two-way ANOVA followed by Turkey's post-hoc test **c** and unpaired two-tailed Student's *t*-test **e**.

might contribute to stimulated lipolysis, probably, through a different mode of action.

## Discussion

In fasted animals, the hydrolysis of stored lipid metabolites needs to be precisely regulated for efficient energy production. As futile fatty acid production resulting from uncontrolled lipolysis is associated with metabolic diseases including obesity and diabetes, lipase activity has to be tightly fine-tuned in response to nutritional status. However, it is not completely understood how lipolysis is spatially and temporally modulated during fasting. In this study, we found that peroxisomes and peroxisomal protein PEX5 could mediate fasting-induced lipolysis by modulating ATGL translocation onto LDs. It is of interest to note that the interaction between PER–LD was increased upon fasting cues to provoke lipolysis. In addition, we found that PEX5 could recognize and escort ATGL to contact sites between PER–LD in a PKA signaling-dependent manner. These findings suggest that peroxisomes and peroxisomal protein PEX5 would play crucial roles in the regulation of energy homeostasis in response to nutritional status (Fig. 8).

LDs appear to contact with other subcellular organelles to mediate lipid homeostasis. As lipolysis takes place on the surface of LDs, it is reasonable to speculate that lipolysis might be affected by subcellular organelles interacting with LD. In this study, we observed that peroxisomes rapidly moved toward LDs upon fasting signals to facilitate lipolysis. In adipocytes, WY promoted peroxisome–LD interaction by increasing the PMP70-positive peroxisomes, as well as the expression of lipolysis-related genes including *Atgl*, leading to further enhancement of ISO-stimulated lipolysis. On the contrary, when the peroxisome movement was inhibited by nocodazole inhibitor or KIFC3 suppression, ATGL translocation onto LDs was blocked in response to fasting cues, eventually reduction of stimulated lipolysis. These findings suggest that the physical contact between PER–LD is sensitively upregulated and modulated by fasting stimuli. On the other hand, we observed that mitochondria, another important organelle for fatty acid oxidation, did not rapidly migrate to LDs during nutritional deprivation. It has been recently reported that LD-interacting mitochondria attenuate fatty acid oxidation and support LD expansion[44]. Thus, it is feasible to speculate that peroxisomes, rather than mitochondria, would quickly respond to fasting signals to initiate catabolic pathways. Although physiological significance of physical interaction between PER–LD in lipid metabolism needs to be further investigated, our findings evidently suggest that the crosstalk between PER–LD would be an important node in lipolysis initiation upon fasting signals.

In the basal state, ATGL is present in the cytoplasm as well as on the LD surface[4]. To trigger prompt and fine-tuned lipolysis, cytosolic ATGL needs to be translocated onto LDs upon fasting stimuli. While it has been shown that ATGL translocates to LDs[10,45–48], the underlying mechanisms by which ATGL migrates to LDs are largely unknown. For the recruitment of ATGL at the contact sites between PER–LD, it seems that certain mediator(s) might be required to escort ATGL. In this study, we have shown that peroxisomal protein PEX5 has an evolutionarily conserved role in peroxisome-mediated ATGL translocation onto LDs upon fasting. In *C. elegans*, PRX-5 suppression via RNAi restored intestinal lipid contents in ATGL-1-overexpressing transgenic worms, implying that PRX-5 might be involved in ATGL-1 dependent lipolysis. Also, in adipocytes, PEX5 associated with ATGL and the interaction was elevated by fasting signals, such as PKA activation. Consistent with the data from worm study, suppression of mammalian PEX5 attenuated the translocation of ATGL onto LDs upon fasting signals, leading to a decrease in stimulated lipolysis. Furthermore, the recruitment of ATGL onto LDs during nutritional deprivation was defective in PEX5 AKO mice. Consistent with our findings, it has been reported that aP2-Pex5 KO mice reduce the level of FFA in WAT under starvation condition[49]. Together, these data clearly suggest that PEX5 can escort ATGL onto LDs at the contact site between PER–LD during fasting. As it has been reported that PEX5 is able to translocate target proteins independently of the canonical PTS1[50,51], it needs to be elucidated how PEX5 can recognize and translocate ATGL upon fasting via a PTS1-independent manner.

Our findings revealed that PEX5 mediated ATGL translocation to contact points between PER–LD upon fasting. This translocation of ATGL by PEX5 seems to require the formation of peroxisome–LD interaction through the migration of peroxisomes to LDs. In this study, we observed that several genes for peroxisomal biogenesis and import appeared to contribute to fasting-induced lipolysis in *C. elegans* and adipocytes (Fig. 4a and Supplementary Fig. 7). For instance, PEX1 suppression attenuated ISO-stimulated lipolysis (Supplementary Fig. 7a and b). In addition, it has been reported that suppression of PEX3 or PEX19 causes peroxisome deficiency[52–54], suppression of PEX19 disrupted translocation of ATGL–PEX5 complex onto LD surfaces in adipocytes (Supplementary Fig. 7c). Also, lipolytic activity was attenuated by suppression of PEX14, an initial peroxisomal docking site for PEX5. These data propose that the interaction between PER–LD would be required for the translocation of ATGL onto LDs by PEX5 in the regulation of fasting-induced lipolysis.

CGI-58 acts as a coactivator of ATGL to mediate stimulated lipolysis[8], and the colocalization of ATGL and CGI-58 is increased by lipolytic stimuli[10]. However, it has been recently shown that

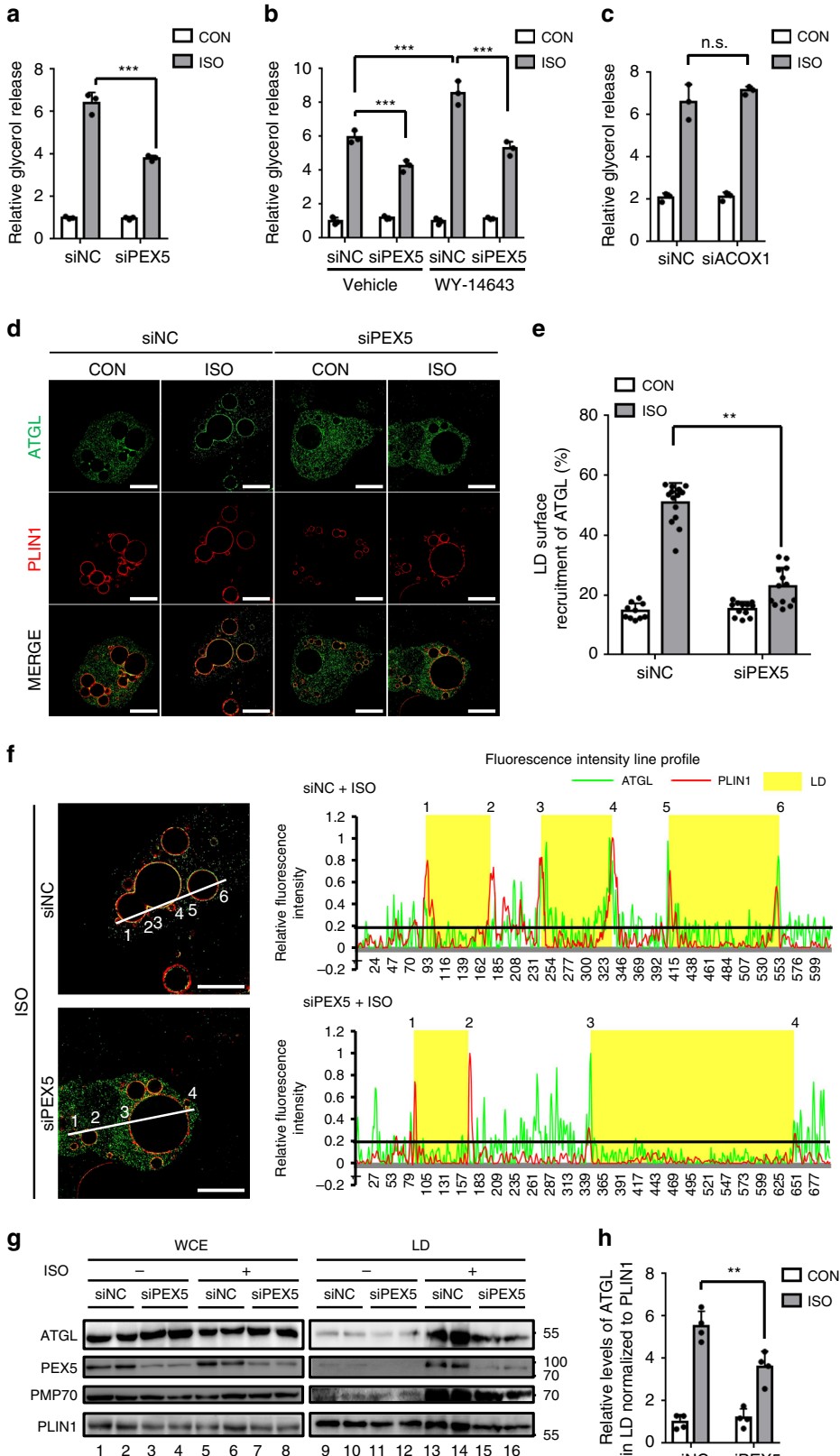

stimulated lipolysis is not completely blocked in CGI-58-deficient mice[55]. Although CGI-58 is important for the hydrolase activity of ATGL, it appears that CGI-58 does not confer whole lipase activity of ATGL in fasting-induced lipolysis. Thus, it is plausible to assume that there might be other pathway(s) to promote ATGL during fasting-induced lipolysis. Here, we showed that PEX5 and

CGI-58 appeared independently to contribute to stimulated lipolysis with ATGL. Although further studies are required, it seems that ATGL would be sequentially regulated by two proteins, PEX5 and CGI-58; PEX5 ushers ATGL to LDs and subsequently, CGI-58 stimulates the activity of the recruited ATGL to boost fasting-induced lipolysis. Based on these data, we propose a novel

**Fig. 5 PEX5 escorts ATGL to LD to mediate fasting-induced lipolysis. a** Relative glycerol release from adipocytes transfected with negative control (NC) or *PEX5* siRNA(siPEX5) for 48 h. $n = 3$ for each group. **b** Relative glycerol release from adipocytes transfected with siNC or siPEX5 for 48 h together in the absence or presence of WY-14643 (10 μM) treatment. $n = 3$ for each group. **c** Relative glycerol release from adipocytes transfected with siNC or siACOX1 for 48 h. $n = 3$ for each group. **d, e** Representative SIM images and quantification analysis of recruited ATGL to LDs in adipocytes immunostained with endogenous PLIN1 (red) and ATGL (green). Cells were transfected with siNC or siPEX5. $n = 10$ cells for siNC group; $n = 15$ cells treated with ISO; $n = 12$ cells for siPEX5 group; $n = 13$ cells for siPEX5 treated with ISO. Quantification of ATGL recruitment to LDs was measured using imageJ software. **f** Representative SIM z-section images (left) and fluorescence intensity profiles from the indicated line scans (right). Below 0.2 fluorescence intensity indicates background fluorescence signal. LD areas are highlighted in yellow. **g** Western blot of whole cell extracts (WCE) or LD fractionation from adipocytes transfected with siNC or siPEX5. 30 μg of protein from WCE; 20 μg of protein from LD fraction. **h** Quantification of ATGL in LDs normalized to PLIN1 from **g**. $n = 4$ independent experiments. CON control, ISO isoproterenol. Cells were treated with ISO (1 μM) for 1 h. All scale bars, 10 μm. Data represent the mean ± SD; *$P < 0.05$, ***$P < 0.01$ in two-way ANOVA followed by Turkey's post-hoc test. n.s., not statistically significant.

pathway in which PEX5 spatiotemporally regulates ATGL-mediated lipolysis at the contact points between PER–LD, probably, independent of CGI-58.

Dysregulation of lipolysis causes several metabolic diseases[56–58]. For instance, ATGL deficiency leads to neutral lipid storage disease (NLSD), which is an autosomal recessive disorder characterized by abnormal cytosolic TG accumulation in multiple tissues[59,60]. In this study, we discovered that PEX5 was required for ATGL-mediated lipolysis. Furthermore, mRNA level of human *PEX5* gene expression was closely linked with that of lipolytic genes, including *ATGL*, in human GTEx analysis, implying that PEX5 may be associated with metabolic diseases, such as lipolysis dysregulation. Consistently, it has been reported that liver-specific PEX5 deletion induces hepatic steatosis due to lipid accumulation, one of the symptoms of NLSD[61]. Therefore, it seems that PEX5-mediated lipolysis might be a potential therapeutic target to treat human metabolic diseases involved in lipolysis dysregulation, such as NLSD.

In conclusion, we showed that the peroxisomal protein PEX5 could mediate fasting-induced lipolysis by recruiting ATGL onto LD surfaces upon fasting stimuli. For an efficient and immediate response to meet energy demands upon fasting cues in animals, it is likely that the interaction between PER–LD is rapidly increased through migration of the peroxisomes to LDs, which is mediated by microtubules and KIFC3, and PEX5–ATGL complex translocates to the contact points between PER–LD (Fig. 8). Taken together, our data suggest that peroxisomes and PEX5 would form an axis for lipid homeostasis in response to fasting signals, which might provide clues for the development of therapies against metabolic diseases associated with lipid dysregulation.

## Methods

**Mice**. In C57BL/6 background, PEX5 AKO mice were generated by crossing adiponectin Cre mice with PEX5-loxP mice (provided by Dr. Baes, KU Leuven, Leuven, Belgium). C57BL/6 male mice were purchased from SAMTACO (Seoul, South Korea) and were housed in colony cages. All experiments with mice were approved by the Seoul National University Institutional Animal Care and Use Committee (SNUIACUC).

**C. elegans strains**. Worms of all strains were grown at 20 °C on nematode growth medium plates seeded with *E. coli* OP50. N2, Bristol was used as the wild-type strain. *hjIs67[atgl-1p::atgl-1::gfp]* was provided by the *Caenorhabditis* Genetics Center (CGC). For feeding RNAi, RNAi clones were obtained from the Ahringer and Vidal RNAi libraries. Synchronized worms were cultured on RNAi plates until they reached the young adult stage.

**Cell lines**. Differentiation of 3T3-L1 adipocytes were induced with DMEM containing 10% fetal bovine serum (FBS), 3-isobutyl-1-methylxanthine (520 μM), dexamethasone (1 μM), and insulin (167 nM) for 48 h. The differentiation induction medium was replaced with DMEM with 10% FBS and insulin (167 nM). Then, medium was changed with DMEM containing 10% FBS every 2 days for 5–7 days. For transfection of siRNAs, differentiated 3T3-L1 adipocytes were washed out and treated with trypsin/EDTA. Then, cells were collected. After washing with PBS, cells were mixed with siRNA and transfected with a single pulse of 1100 V for 30 ms using a Microporator MP-100 (Digital Bio). HEK293T cells (ATCC, CRL3216) were transiently transfected with various DNA plasmids using the calciumphosphate method.

**Lipid staining**. *C. elegans* was harvested in 60 μl of PBS, 120 μl of 2 MRWB buffer (40 mM NaCl, 160 mM KCl, 14 mM Na2-EGTA, 0.4 mM spermine, 1 mM spermidine–HCl, 30 mM Na–PIPES (pH 7.4), 0.2% β-mercaptoethanol), and 60 μl of 4% paraformaldehyde. *C. elegans* was freeze-thawed three times and washed with PBS. Then, *C. elegans* was dehydrated in 60% isopropyl alcohol for 10 min at room temperature and stained with ORO solution. To visualize LDs in 3T3-L1 adipocytes, cells were fixed with 1% paraformaldehyde. BODIPY 493/503 stock solution (D-3922; Molecular Probes; 1 mg ml$^{-1}$ in methanol) was diluted 1:1000 in PBS before being added to and incubated with cells for 30 min. Mounting medium (Vectashield without DAPI; Vector Laboratories) was used for fluorescence imaging. Adipocytes were observed and imaged using an LSM 700 confocal microscope (Zeiss) and an OMX SIM microscope (GE Healthcare Life Sciences).

**Immunofluorescence microscopy**. Cells were plated on coverslips (Marienfeld precision cover glasses 1.5 H) fixed in 4% paraformaldehyde for 10 min, and permeabilized with 0.5% TritonX-100 in PBS for 5 min. In the case of digitonin treatment to decrease cytosolic staining, cells were treated with PBS containing 0.01% digitonin for 5 min on ice, followed by fixation with 4% paraformaldehyde, as described previously[45]. Fixed cells were incubated with primary antibodies against PLIN1 (20R-pp004; Fitzgerald Industries, 1:300), ATGL (2138S; Cell Signaling, 1:300), PMP70 (ab3421; Abcam, 1:300 (all fluorescence images of immunostained PMP70 except for Supplementary Fig. 4d), SAB4200181; Sigma, 1:300 (fluorescence images of immunostained PMP70 for Supplementary Fig. 4d), PEX5 (ab125689; Abcam, 1:300 (fluorescence images of immunostained PMP70 for Fig. 6a), sc-23188; Santacruz, 1:200 (fluorescence images of immunostained PMP70 for Supplementary Fig. 4d)), Catalase (ab16771, Abcam, 1:300) overnight, washed three times for 5 min each, incubated in secondary antibody for 1–2 h, washed three times for 5 min each, and mounted on glass slides with mounting medium (Vectashield without DAPI; Vector Laboratories). Cells were observed and imaged using an LSM 700 confocal microscope, and an OMX SIM microscope.

**Live cell imaging**. Cells used for live cell imaging were maintained at 37 °C in the presence of 10% $CO_2$ in the OMX microscope. *C. elegans* used for live imaging were treated with 1 mg ml$^{-1}$ levamisole to immobilize *C. elegans* for 4 h. *C. elegans* was observed under a CARS microscope (Leica) and the confocal microscope (Zeiss).

**Quantification of microscopic images**. *Lipid contents*: CARS microscopy was used to visualize and quantify lipid contents in *C. elegans* without any staining[62,63]. The intensity of visualized lipid contents (CARS intensity) was quantified using intrinsic vibrational contrast of lipids according to CARS manufacturer's instructions (Leica). CARS intensity was measured by LAS X (Leica software).

*Colocalization between PER–LD*: The level of PMP70 protein was quantified by sum of PMP70 fluorescence intensities of each z-section using imageJ software. Contact between PER–LD was defined as the colocalization between PER–LD. Peroxisomes were immunostained with PMP70 or overexpressed fluorescence protein DS-RED-PTS, and LDs were stained with PLIN1 or BODIPY. Colocalization signals between PER–LD were marked by double positive fluorescence signals between PER–LD from scatter plot using microscopic software (Zen, Zeiss; LAS X, Leica; Softworx, GE healthcare Life Sciences). Then, the pixel fluorescence intensity of marked colocalization was quantified using imageJ software. Colocalization ratio was calculated by the fluorescence intensity of peroxisomes colocalized with LDs divided by total fluorescence intensity of peroxisomes (colocalization ratio = the fluorescence intensity of peroxisomes colocalized with fluorescence dye of LDs/total fluorescence intensity of peroxisomes).

*ATGL recruitment onto LD surface*: Percentage of ATGL recruitment onto LD surface was calculated by the number of ATGL signals in close proximity (<0.2 μm) within LD surface divided by total number of ATGL signals using imageJ software[24].

*The size of peroxisomes*: The size of peroxisomes was categorized by pixel-based classification using LSA X. Pixel units were converted to μm units. Then,

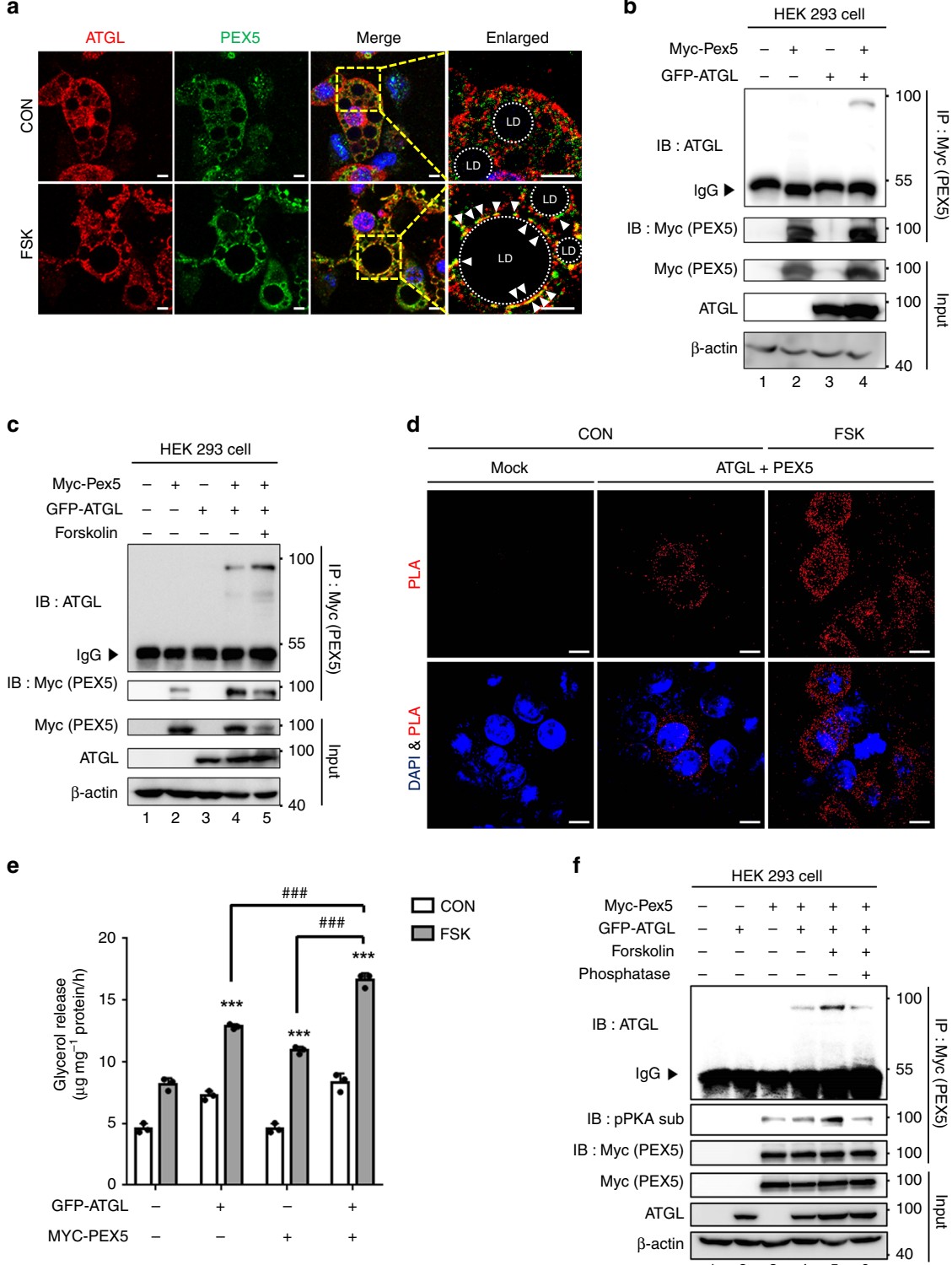

**Fig. 6 PEX5 mediates stimulated lipolysis through interaction with ATGL. a** Representative images of differentiated adipocytes immunostained for endogenous ATGL (red), PEX5 (green), and DAPI (blue). Adipocytes were treated with or without FSK (10 µM) for 1 h. CON control; FSK forskolin. Scale bars, 10 µm. **b, c** Co-immunoprecipitation with an anti-MYC antibody and western blotting were conducted with the indicated antibodies. HEK293 cells were transfected with MYC-PEX5 and GFP-ATGL expression vectors. FSK forskolin. **d** In HEK293 cells, in situ proximity ligation (PLA) assay transfected with MYC-PEX5 and GFP-ATGL expression vectors without lipid challenge. FSK forskolin. Scale bars, 10 µm. **e** Concentrations of released glycerol from cultured COS-7 cells transfected with or without PEX5-WT and ATGL-WT expression vectors. COS-7 cells were pretreated with oleic acid (500 µM) for 48 h and FSK (25 µM) for 3 h. $n = 3$ for each group. CON control; FSK forskolin. Data represent the mean ± SD; ***$P < 0.001$ vs. Mock-CON and ###$P < 0.001$ group by two-way ANOVA followed by Turkey's post-hoc test. **f** HEK293 cells were transfected with MYC-PEX5 and GFP-ATGL expression vectors and treated with or without FSK and phosphatase (CIAP). Co-immunoprecipitation with an anti-MYC antibody and western blotting were conducted with the indicated antibodies. IP immunoprecipitation; IB immunoblotting; IgG immunoglobulin G.

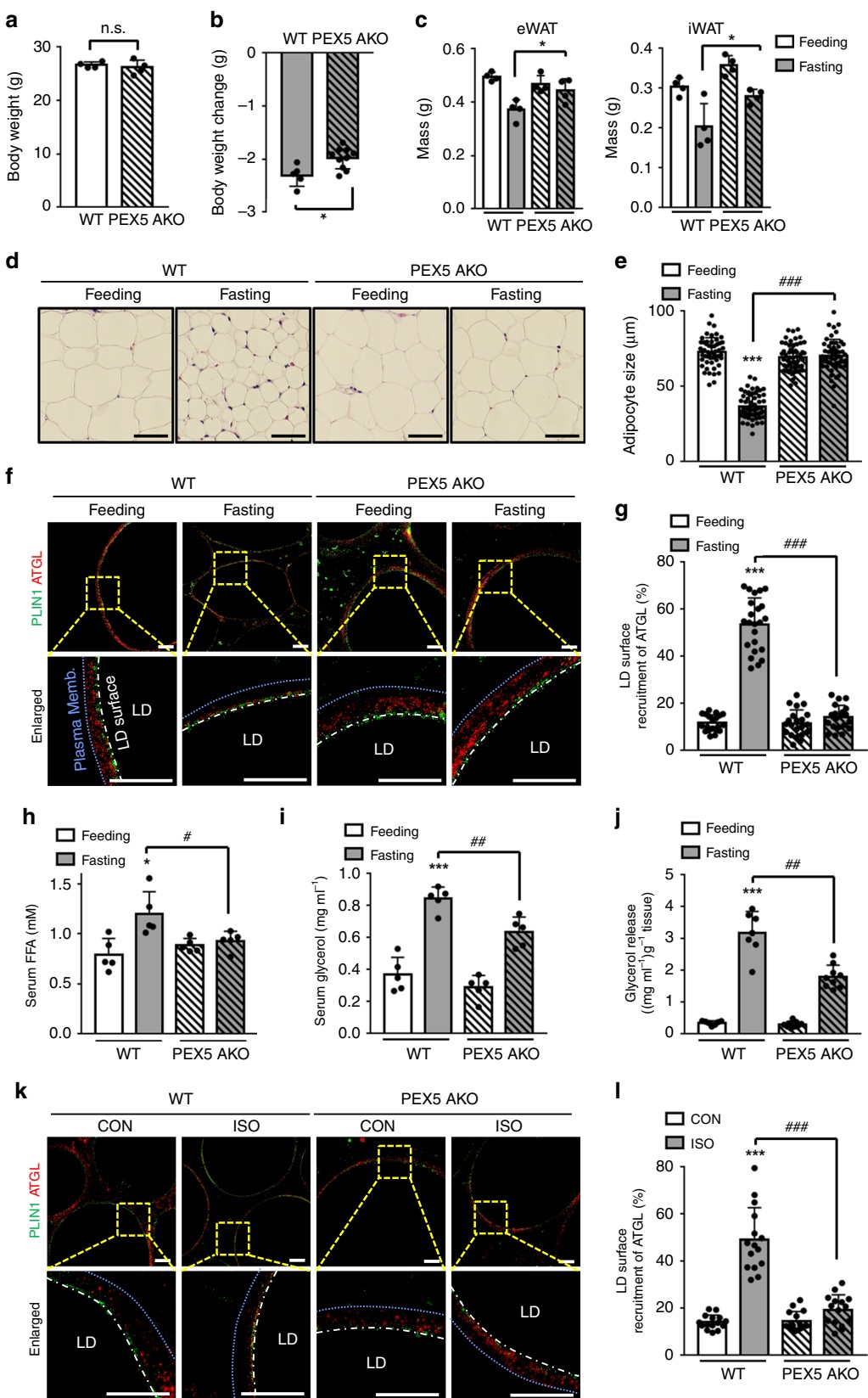

peroxisomes were classified from 0.05 to 0.65 μm² into 0.05 intervals in 3T3-L1 adipocytes (from 0.05 to 0.6 μm² into 0.05 intervals in *C. elegans*).

**In situ PLA**. Paraformaldehyde-fixed cells were washed with PBS and blocked with blocking solution. The primary rabbit antibody was applied, and the cells were

incubated with plus and minus secondary PLA probes against both rabbit and mouse IgG. The incubation was followed by hybridization and ligation, and then amplification was performed. After mounting with DuoLink-mounting medium, the samples were examined using an OMX SIM microscope and LSM 700 confocal microscope.

**Fig. 7 Fasting-induced lipolysis is attenuated in PEX5 AKO mice. a** Body weight of WT and PEX5 AKO mice (12 weeks). $n = 4$ per genotype. **b** Body weight changes of WT and PEX5 AKO mice under fasting (12 h) conditions. $n = 5$ for WT; $n = 10$ for PEX5 AKO. **c** Masses of eWAT and iWAT from WT and PEX5 AKO under feeding and fasting (12 h) conditions. $n = 4$ for each group. **d** Adipocyte morphology of eWAT from WT and PEX5 AKO mice stained by hematoxylin and eosin (H&E). Scale bars, 50 μm. **e** Adipocyte size in eWAT of WT and PEX5 AKO mice under feeding and fasting (12 h) conditions. $n = 52$ for feeding WT; $n = 61$ for fasting WT; $n = 54$ for feeding PEX5 AKO; $n = 53$ for fasting PEX5 AKO. **f** Representative SIM images showing the subcellular localization of ATGL in eWAT immunostained with PLIN1 (green) and ATGL (red) under feeding and 12 h of fasting. Blue dashed line, plasma membrane; white dashed line, LD surface. The boundaries of adipocytes membrane was distinguished by DIC images. Scale bars, 10 μm. **g** Quantification of ATGL recruitment to LDs in eWAT of **f**. $n = 20$–23. **h, i** Serum levels of FFAs **h** and glycerol **i** in fed and fasted (12 h) mice. $n = 5$ per genotype. **j** Ex vivo lipolysis measured by glycerol release from eWAT treated with or without ISO (5 μM) for 1 h. $n = 7$ for WT; $n = 9$ for PEX5 AKO. **k** Representative SIM images of ATGL recruitment in eWAT immunostained with PLIN1 and ATGL treated with or without for ISO (5 μM) for 1 h. Scale bars, 10 μm. **l** Quantification of ATGL recruitment to LDs in eWAT of **k**. $n = 15$ per genotype. Data represent the mean ± SD; *$P < 0.05$, ***$P < 0.001$, vs. WT-Feeding or WT-CON, #$P < 0.05$, ##$P < 0.01$, and ###$P < 0.001$ group in two-way ANOVA followed by Turkey's post-hoc test.

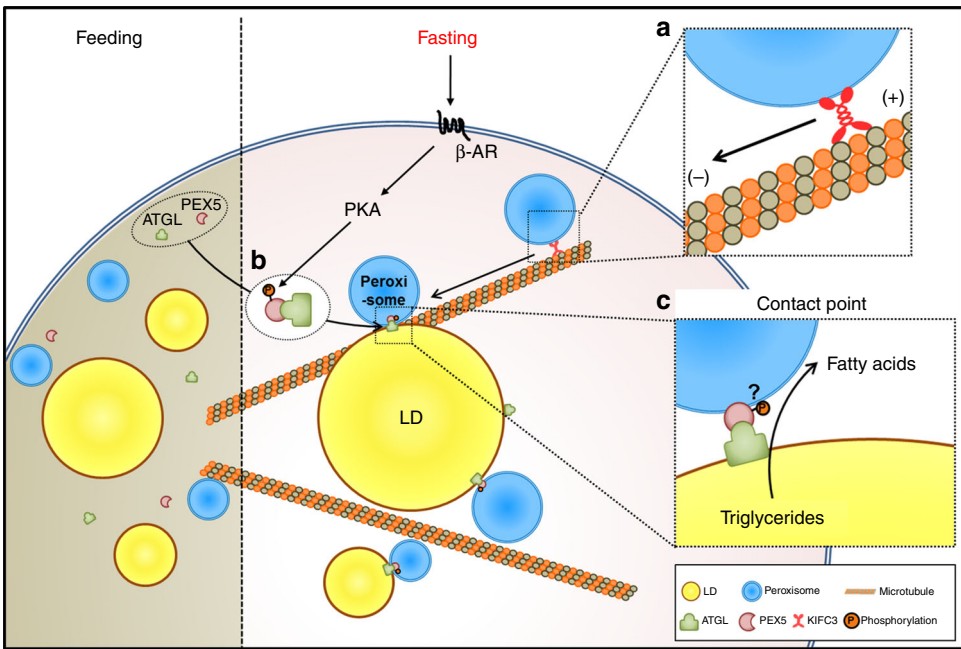

**Fig. 8 Working model.** During fasting, peroxisomes move to LDs via microtubule and this movement is mediated by KIFC3 (**a**). PKA activation increases PEX5 phosphorylation and PEX5 interacts with ATGL independent of canonical PTS1 (**b**). Then, PEX5–ATGL complex translocates to the contact points between PER–LD even though it needs to be further elucidated whether PEX5 phosphorylation is important for PEX5–ATGL translocation to the contact points between PER–LD. Finally, fasting-induced lipolysis is elevated (**c**).

**Peroxisome protection assay.** The crude peroxisomal fraction was isolated using the Peroxisome Isolation Kit (PEROX1, Sigma). The fractionation samples were treated with or without Proteinase K (25530049, invitrogen) 200 μg ml$^{-1}$ and 1% Triton X-100. Both groups were incubated on ice for 30 min. Samples were processed by western blot assay.

**Lipid droplet isolation.** 3T3-L1 adipocytes were collected and suspended in buffer A (20 mM tricine and 250 mM sucrose, pH 7.8) with 0.2 mM PMSF and kept on ice for 20 min. Then, the sample of 3T3-L1 adipocytes was homogenized on ice and postnuclear supernatant (PNS) fraction was obtained by centrifugation at $3000 \times g$ for 10 min at 4 °C by removing nuclei, cell debris, and unbroken cells. 10 ml of the PNS fraction was transferred to SW40 tube and then 2 ml of buffer B (20 mM HEPES, 100 mM KCl, and 2 mM MgCl$_2$, pH 7.4) was loaded on top of the PNS. The sample was centrifuged at $100,000 \times g$ for 2 h at 4 °C using an ultracentrifuge. Then, LD fraction was obtained from the top band of the gradient.

**Cell lysis and immunoprecipitation.** After washing with PBS, cells were treated with TGN buffer (50 mM Tris–HCl, pH 7.5, 150 mM NaCl, 1% Tween-20, and 0.3% NP-40) supplemented with 0.1% protease inhibitor cocktail (Roche, Basel, Switzerland). Total cell lysates were obtained by centrifugation at $13,000 \times g$ for 15 min at 4 °C, and 1–1.5 mg of lysates was used for immunoprecipitation. The lysates were incubated with primary antibodies for 2 h at 4 °C, followed by a 1-h incubation with 50% slurry of protein A sepharose presaturated with the lysis buffer. After washing three times with lysis buffer, the immunoprecipitated proteins were recovered from the beads by boiling for 10 min in sample buffer and were analyzed by SDS–PAGE and immunoblotting.

**Western blotting.** Cells were lysed on ice with modified RIPA buffer containing 50 mM Tris–HCl (pH 7.5), 150 mM NaCl, 2 mM EDTA, 1% (v/v) Triton X-100, 0.5% (w/v) sodium deoxycholate, 0.1% (w/v) SDS, 5 mM NaF, 1 mM Na$_3$VO$_4$, and a protease inhibitor cocktail (GeneDEPOT; #P3100). Antibodies against ATGL (2138S, RRID:AB_2167955, Cell Signaling, 1:1000), HSL (4107S, RRID: AB_2296900, Cell Signaling, 1:1,000), pPKA substrate (9624S, RRID:AB_10692481, Cell Signaling, 1:1000), PLIN1 (20R-pp004, Fitzgerald Industries, 1:1000), PEX5 (NBP2-38443, Novus biological, 1:1000), CATALSE (ab16771, Abcam, 1:1000), MYC (9B11, Cell signaling, 1:1000), GAPDH (LabFrontier, Co., LF-PA0018, 1:1000), and β-actin (Sigma, A5316, RRID:AB_476743, 1:2000) were used for western blotting analysis. Original full immunoblots were shown in Supplementary Fig. 8.

**Glycerol release and FFA release assay.** Glycerol release from adipocytes or adipose tissues was measured using free glycerol reagent (F6428; Sigma), FFA reagent (11383175001; Roche) according to the manufacturer's protocol. For lipolysis ex vivo, eWAT isolated from mice were cut into 50-mg pieces and incubated at 37 °C in DMEM containing 2% FA-free BSA.

**Serum profiling.** Serum FFA and glycerol levels were assessed using FFA (Roche), TG (Thermo Fischer Scientific).

**Quantitative reverse transcription (qRT)-PCR.** Total RNA was isolated from adipocytes. cDNA was synthesized using a reverse transcriptase kit (Thermo Fisher Scientific) according to the manufacturer's instructions. Primers used for qRT-PCR

were obtained from Bioneer (South Korea). The primer sequences used for the real-time quantitative PCR analyses are described in the Supplementary Table 1.

**Statistical analysis**. All data were analyzed using Student's $t$-test or analysis of variance (ANOVA) in Excel (Microsoft) or GraphPad Prism7; $p$ values of < 0.05 were considered significant.

**Reporting summary**. Further information on research design is available in the Nature Research Reporting Summary linked to this article.

## Data availability

The data supporting the findings of this study are available from the corresponding author upon reasonable request.

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

## Acknowledgements
This work was supported by the National Creative Research Initiative Program of the National Research Foundation (NRF) funded by the Korea government (Ministry of Science, ICT and Future Planning; 2011-0018312). J.K., Y.J., Y.G.J., J.S.H., J.H.L., G.L. and H.J. were supported by the BK21 program.

## Author contributions
J.K. designed and conducted the study, performed experiments, and wrote the manuscript. Y.J., Y.G.J., J.S.H., K.H.H., J.H.L., G.L., H.J. and S.S.C. performed experiments and contributed to the writing of the manuscript. M.B. has contributed to interpreting the data and discussing whole project. J.B.K. supervised the whole project, discussed the data, and edited the final manuscript.

## Competing interests
The authors declare no competing interests.
