## [Peer Review File · Nature Communications]

Reviewers' comments:

Reviewer #1 (Remarks to the Author):

The study of Kong et al provides evidence for a regulatory role of peroxisomes in stimulated lipolysis. The authors demonstrate that peroxisomal marker proteins co-localize with lipid droplets (LDs) and are involved in the translocation process of ATGL from the cytoplasm to LD. Evidence (although a bit preliminary) is also provided for a direct interaction between peroxisomal PEX5 and ATGL mediating enzyme translocation. Overall the observations are interesting and new. However, some of the presented data are preliminary and over-interpreted. Additional experiments are required to corroborate the conclusions of the study.

Specific points

1. Analysis of lipolysis should additionally include data for fatty acid (FA) release. Glycerol release alone is not sufficient to evaluate the activity of ATGL.
2. Data on ATGL translocation to LDs are not very consistent throughout the manuscript. The claim that the enzyme is mostly localized on LDs (page 10) is not supported by Western blotting analyses in Figs. 1I, 5A and 6A. Generally, single bands on Western blots are not very informative. Double or triple sample application would be more convincing.
3. Translocation of p-HSL to LDs is much more pronounced than ATGL translocation (Fig. 1I). Does HSL translocation also depend on PEX5 or peroxisomes? Clarification of this question is essential for this study!!!
4. Blockade of microtubule polymerization by nocodazole or knockdown of KIFC3 kinesin may not be specific for peroxisome movement but also affect other cellular transport processes. Accordingly, the presented experiments require a negative control to determine the peroxisome-independent effect of nocodazole and KIFC3.
5. KIFC3 kinesin knock down efficiency should be documented by either western blot or qRT-PCR.
6. The conclusion that peroxisome proliferation in response to WY14643 affects lipolysis may be incorrect because WY not only affects peroxisome abundance but also ATGL transcription. Accordingly the results displayed in Fig. 2g-i are not very informative.
7. The description of lipolytic parameters in PEX5-AKO mice is quite preliminary and incomplete. Total adipose mass and the masses of specific adipose depots should be presented. Also, the contribution of ATGL and HSL to FA and glycerol release should be determined using specific enzyme inhibitors. Potential lipid accumulation in other organs such as liver and heart may be indicative of a potential regulatory role of peroxisome for lipolysis in non-adipose tissues! What about LD localization of ATGL and PMP70 in PEX5-AKO mice???
8. Interaction studies between ATGL and PEX5 are preliminary and require further experimental proof by proximity ligation assays or FRET.
9. The effect of PEX5 silencing in the absence of CGI-58 on lipolysis seems rather minor. Accordingly, the conclusion that PEX 5 regulates lipolysis independently of CGI-58 requires better experimental support.
10. Fig. 6d lacks an experiment GFP-ATGL+; Myc-PEX5- in the presence or absence of forskolin. Without is the claim that ATGL activity depends on PEX5 is not justified.

Reviewer #2 (Remarks to the Author):

The authors found that physical interaction between peroxisomes and lipid droplets (LDs) occurs in differentiated 3T-L1 adipocytes upon fasting, which stimulates glycerol release by enhancing lipolysis. The author also showed that PEX5 interacts with adipose triacylglycerol lipase (ATGL), a rate-limiting enzyme for TG lipolysis, and efficiently colocalizes with it on LDs in the presence of fasting cues. Furthermore, the recruitment of ATGL to LDs was impaired in adipocytes transfected siRNA against PEX5 as well as adipocytes derived from adipose-specific PEX5-knockout mice, resulted in attenuation of fasting-induced lipolysis. From these results, the authors proposed a working model that upon fasting PEX5 recognizes and binds to ATGL, the resultant PEX5-ATGL complex translocates to the contact points (so to speak "contact sites"?) between peroxisomes and LDs, eventually leading to fasting-induced lipolysis. These findings are interesting and important to uncover the molecular mechanism of tightly regulated lipolysis in adipocytes. However, translocation of PEX5-ATGL complex to the contact points between peroxisomes and LDs was not clearly shown. In addition, there are several critical issues that need to be addressed as summarized below.

Major comments

1. The authors showed that physical contacts of peroxisomes with LDs were augmented upon fasting conditions, which was correlated with fasting-induced lipolysis (Figs 1 and 2). However, it is not clear whether physical contacts between peroxisomes and LDs were essential for lipolysis upon fasting or just a response to fasting. Indeed, peroxisome itself appears to be dispensable for fasting-induced lipolysis as assessed by normal lipolysis in the *PEX19* (*prx-19*) RNAi-treated worms (Fig. 4a). Are physical contacts between peroxisomes and LDs required for fasting-induced lipolysis and accumulation of ATGL and PEX5 on LDs? Please explain them with direct evidence including localization of ATGL and PEX5 in *PEX19*-siRNA-treated adipocytes upon fasting.
2. The authors found that translocation of peroxisomes but not mitochondria to LDs upon fasting and treatment with ISO or forskolin. Although the physical contacts between peroxisomes and LDs were shown by the presence of PMP70, a peroxisomal membrane protein, in the fractions enriched in LDs (Fig. 1i), distribution of peroxisomal matrix and mitochondrial proteins should be shown to strengthen the data in Fig. 1b and Supplemental Fig. 1d.
3. The authors claimed that PEX5 escorts ATGL to the contact points between peroxisomes and LDs in the presence of fasting cues. However, the localization of ATGL and PEX5 on the contact points between peroxisome and LDs was not directly shown, though ATGL was largely colocalized with PEX5 on LD upon forskolin treatment (Fig. 6a). This must be demonstrated.
4. By suppressing the expression of peroxisomal protein(s) using RNAi, the authors found that suppression of *PEX5* (*prx-5*) but not other *PEX* (*prx*) genes attenuated LD hydrolysis upon fasting in *C. elegans* (Fig. 4a). However, it is expected that several functions of peroxisome be compromised by the knockdown of *PEX5* as well as other *PEX* genes. So that what is the physiological consequence of the formation of contact between peroxisomes and LDs? Is there any specificity in the fatty acids of triacylglycerol during lipolysis upon fasting?
5. Fig. 6: The authors showed the interaction between ATGL and PEX5 by immunoprecipitation assays. Do they interact directly and how does PEX5 interact with ATGL harboring no typical peroxisome-targeting signal type 1 (PTS1)? And an equally important issue including mechanistic insight of the PEX5-mediated localization of ATGL to the contact points between peroxisomes and LDs should be shown.
6. Fig. 6e: Phosphorylation of MycPEX5 was demonstrated by immunoblot using pPKA antibody that detects peptides and proteins containing a phospho-serine/threonine residue with arginine at the -3 position. Is there any putative phosphorylation site(s) in PEX5? Phosphorylated PEX5 upon PKA activation with forskolin and the reduced level of phosphorylated PEX5 upon phosphatase treatment should be demonstrated. Why was the signal corresponding to the PKA-mediated phosphorylated PEX5 in lane 3 higher than that in the case of both expressed together with GFP-

ATGL in lane 4?

7. Fig. 8: There was no result demonstrating that either ATGL or PEX5 is recruited to the contact points between peroxisomes and LDs. In addition, dephosphorylation of PEX5 that had been recruited on the contact points between peroxisomes and LDs was not shown. Are PEX5 and/or ATGL essential for the formation of contact points between peroxisomes and LDs? These issues are critical to review the authors' working model.

8. In Figs. 3a and 3b, ATGL-1 in fasting worm and ATGL in ISO-treated adipocytes were localized to peroxisomes in the cytoplasm in addition to the peroxisomes in proximity to LDs. Why?

9. Furthermore, PEX5 was partly present in LD fraction even in normal condition (Fig. 5g). Given the findings that localization of PEX5 to LDs appears to be more efficient than that of PMP70 (peroxisomes) upon fasting (Fig. 2 and 6a) and PEX5 is specifically involved in fasting-induced lipolysis in *C. elegans* (Fig. 4a-d), regulation of fast-induced lipolysis in adipocytes might be explained as a novel function of PEX5, independent of physical contacts between peroxisomes and LDs. The authors need to clearly distinguish between the roles of peroxisome-LD contacts and PEX5 in fasting-induced lipolysis.

10. Import of peroxisomal matrix proteins in *C. elegans* completely depends on PTS1 due to the lack of PTS2 pathway. So, the roles of PEX5 in the peroxisomal matrix protein import processes between *C. elegans* and mammals are likely distinct. Such issues need to be addressed before explaining and discussing the novel function of PEX5 in fasting-induced lipolysis.

Minor comments

1. Figs. 1i and 5g: Relative amounts of LD fractions to whole cell extracts (WCE) should be shown. Distribution of PMP70 needs to be shown in Fig. 5g. This is critical to validate the efficiency of physical contacts between peroxisomes and LDs.

2. Fig. 3a: Why were most of ATGL-1 co-localized with DS-RED-PTS1 even at the fasting period 0 hour?

3. Fig. 5, a-c: The levels of relative glycerol release by the ISO-treated adipocytes that were transfected with negative control siRNA (siNC, solid bars) are much higher in 5a than in 5b and 5c despite the same experimental condition. Why is that so?

4. Fig. 5c: How much were the protein level of ACOX1 and/or peroxisomal fatty acid oxidation level suppressed in this experimental condition?

5. Fig. 5 h: Statistical analysis is required.

6. Fig. 6, b, c, and e: Size markers should be indicated.

7. Fig. 6d: Expression of only GFP-ATGL should be included as a control to verify the effect of PEX5 in lipolysis.

8. Supplementary Figure 1d: Mitochondrial signal was too weak in the presence of ISO.

9. Supplementary Figure 5h: Statistical analysis is required.

10. Several recent papers reported on the relationship between peroxisomes and LDs (Schrul and Kopito, *Nat. Cell Biol.* 18: 740-751 (2016), Wang et al., *Nat. Commun.* 9: 2939 (2018), Joshi et al., *Nat. Commun.* 9: 2940 (2018)). They should be properly cited and discussed.

Reviewer #3 (Remarks to the Author):

This is an excellent study that enriches the presently limited but growing body of knowledge around the peroxisome's role in lipid metabolism in adipose tissues. Here, Kong et al investigate the mechanism by which peroxisomes and lipid droplets (LD) tether to each other in adipocytes during fasting. Previous studies have shown that peroxisomes may have a role in regulating LD in adipose tissue. Using multiple systems including culture cells, *C. elegans* and mouse models, Kong et al show that the peroxisomal matrix import protein receptor, PEX5, is not only required to form the peroxisome-LD tether but also recruit ATGL to LD during fasting.

The work begins with the demonstration of an increase in peroxisome-LD interaction during fasting in *C. elegans*, mouse epididymal adipose tissue and 3T3-L1 in a microtubule and KIFC3 dependent fashion. Using the same systems, they also show that PEX5, a cytosolic peroxisomal matrix protein receptor, is required to recruit ATGL to LD assumingly to the site where the peroxisome-LD tether is formed; this tether is required for the lipolysis during fasting in adipose tissue. Finally, they present that in adipose tissue from PEX5 KO mice, ATGL recruitment is defective and show reduced lipolysis. While defect in adipose tissue development has previously been demonstrated in these α 2-PEX5 KO mice, the finding here suggests a possible mechanism by which peroxisomes regulate LD in adipose tissues. However, in order to be considered for publication in Nature Comm, the authors need to consider the following points about this study and make appropriate experimental and presentation-related amendments.

- 1) It is not clear where and how ATGL is localized with peroxisomes. The authors make a convincing argument for PEX5-ATGL interaction. However, how is PEX5 recruiting ATGL to LD? Does it required Peroxisomes? If peroxisomes as a whole is required for ATGL recruitment to LD, is PEX5 recruiting ATGL into the matrix or just to the surface? In Figure 3a, ATGL appears to colocalize with DS-RED-PTS even before fasting, suggesting that ATGL is localized to peroxisomes prior to being targeted to LD. Since PEX5 is the matrix protein cytosolic receptor, one would postulate that ATGL would be imported into peroxisomes. If this is the case, how would ATGL act on the LD? However, if PEX5 recruits ATGL to the surface of peroxisomes and LD, then there may be other other cellular sequences such as peroxisome degradation. It has previously shown that any aborted import that results in PEX5 accumulation on the peroxisomal membrane has been shown to induce peroxisome degradation. In other words, if ATGL recruitment requires PEX5 on the peroxisomal membrane as the author states, then this may induce pexophagy due to the accumulation of membrane bound PEX5. It may be difficult to imagine that ATGL would be in the matrix of the peroxisomes as it postulated function is to activation of lipolysis of, however, such a mechanism may be possible if peroxisomes protrudes into LD as shown for *S.cerevisiae*. This is an important point as ATGL localization inside peroxisomes suggests a novel mechanism for lipolysis in LD. Similarly, if PEX5 recruits ATGL to the peroxisomal surface or just to the peroxisomal matrix import pathway, it suggests a novel import pathway that is mediated by a peroxisomal protein. To address the sub-cellular localization, the authors should attempt the protease protection assay in intact cells using agents that semi-permeabilize the plasma membrane such as digitonin.
- 2) Similarly, it is not clear whether peroxisomes themselves are required for PEX5 to recruit ATGL to the site of Peroxisome-LD contact. It may be that PEX5 alone may be necessary. Based on the RNAi screen for peroxisome proteins required for lipolysis in *C. elegans*, it appears that only PEX5 and not its membrane receptors, PEX14 and PEX13, is needed. Does this mean that PEX5 alone is required? This question must be addressed within the scope of this study to clarify the role of PEX5 and/or peroxisomes in LD regulation. To test this, the authors should use peroxisome-deficient cells to determine whether PEX5 alone is necessary for the recruitment of ATGL to LD.
- 3) Using the quantification of total PMP70 fluorescent intensity from a confocal images, the authors suggest that PMP70 does not increase; hence peroxisomes do not increase. However, in Figures 1c and 1d, the peroxisomes during fasting look sizably larger then in fed conditions, and in Fig. 1g, there appears to be more PMP70 in the ISO-treated WCL. Using fluorescent intensity from immunofluorescent images can be problematic and inaccurate. Is it possible that there are more as well as larger peroxisomes in the adipose during fasting? Given that β -adrenergic receptor induces PPAR-alpha, a nuclear receptor that induces peroxisome proliferation, could it be possible that ISO treatment is inducing an increase in peroxisome size and numbers, which causes an increase in peroxisome-LD interaction?
- 4) It is very difficult to assess some of the figures as information is either hidden or missing. For example, there is no explanation for the way the fluorescent intensity of PMP70 was quantified. Similarly, there is no information about the colocalization quantification. Such information is critical for the reader to assess the data.
- 5) Furthermore, the methods section in general needs improvement. The brevity of the method section made it difficult for this reviewer to fully understand the experiments and some

assumptions that have been made. This is especially important as the authors have used multiple model systems and greater clarity is required for readers to understand each of the different systems. For example, what is relative CARS intensity? Another example is the settings for microscopy. While the authors have defined the abbreviations in the main text, providing full descriptions in the methods section would be useful. Greater details are also needed in the figure legend.

6) Does the translocation of peroxisomes to LD require ATGL? This question should be easy to address and would help test whether ATGL and PEX5 form the peroxisome-LD tether.

7) Why is PEX5 in the LD fraction in the non-ISO condition (Fig. 5g) while there is no PMP70 (Fig 1i)? Is it possible that PEX5 targets to LD independently of peroxisomes? If this is the case, it may explain why the depletion of PEX14 does not prevent a reduction of LD as seen with PEX5 knockdown.

8) Fig 6d: what happens to glycerol release levels when expressing ATGL or PEX5 alone? Is the effect (lipolysis stimulation) additive or dependent?

Minor:

- Fig. 1f: Recommend showing the enlarged images with both PLIN1 and PMP70 (instead of PMP70 alone) so that readers can easily see the colocalization between peroxisomes and LDs.
- Fig. 3: Quantification (colocalization) is required for all images.
- Fig. 5e: Define how the measurement works in the figure legend and/or method. What does "Recruitment of ATGL1 %" mean here?
- Fig. 5h: Need to show the number of trials performed and statistical analysis.
- Fig. 6a: Visualization of this figure would be greatly aided with an enlarged region
- It is difficult to observe the movement of peroxisomes in the time lapse imaging in Fig. S1c, even in the movie. Perhaps showing the separate channels along with the merge for each time frame would make it easier to view the movement of the peroxisomes towards the LD. The authors should also consider showing divided images for the movies, but with a slower time rate than currently used.
- Fig. 1b: What does the y-axis "colocalization ratio %" mean? Which parameter is being measured here, intensity, number or area? The authors need to explain it in the figure legend. Also define "white arrowhead". The authors need to explain in detail how the degree of colocalization was measured/calculated for ALL the figures. I have no idea how their "colocalization ratio" was determined.

Response to the Reviewers' Comments

MS ID#: NCOMMS-18-6227380

MS TITLE: Spatiotemporal contact between peroxisomes and lipid droplets regulates fasting-induced lipolysis via PEX5

Reviewers' comments:

Reviewer #1 (Remarks to the Author):

The study of Kong et al provides evidence for a regulatory role of peroxisomes in stimulated lipolysis. The authors demonstrate that peroxisomal marker proteins co-localize with lipid droplets (LDs) and are involved in the translocation process of ATGL from the cytoplasm to LD. Evidence (although a bit preliminary) is also provided for a direct interaction between peroxisomal PEX5 and ATGL mediating enzyme translocation. Overall the observations are interesting and new. However, some of the presented data are preliminary and over-interpreted. Additional experiments are required to corroborate the conclusions of the study.

Specific points

1. Analysis of lipolysis should additionally include data for fatty acid (FA) release. Glycerol release alone is not sufficient to evaluate the activity of ATGL.

According to the reviewer's comment, we measured the levels of FA release. As shown in new Figure 2f, i, Supplementary Figure 4a, 5a and 5b, the tendency of released FAs was similar to that of released glycerol. We described these in the revised manuscript (p. 7, p.10, and p. 13).

2. Data on ATGL translocation to LDs are not very consistent throughout the manuscript. The claim that the enzyme is mostly localized on LDs (page 10) is not supported by Western blotting analyses in Figs. 1I, 5A and 6A. Generally, single bands on Western blots are not very informative. Double or triple sample application would be more convincing.

Following this comment, we have performed new western blot analyses and provided new data. As shown in Figure 1i, 5g and 5h, ATGL translocation to LDs was clearly enhanced upon ISO treatment. We described these in the revised manuscript (p. 11)

3. Translocation of p-HSL to LDs is much more pronounced than ATGL translocation (Fig. 1I). Does HSL translocation also depend on PEX5 or peroxisomes? Clarification of this question is essential for this study!!!

The points are well taken. It has been demonstrated that both ATGL and HSL are important for lipolysis. However, recent studies including ours and others have suggested that ATGL would have key roles for fasting-induced lipolysis, compared to HSL^{1,2}. Moreover, while ATGL deficient mice show enhanced adiposity accompanied with impaired adipocyte lipolysis under both basal and isoproterenol-stimulated conditions, HSL null mice exhibit no alteration of TG accumulation in both

adipose and non-adipose tissues, compared to WT mice^{3,4}. In addition, it has been reported that ATGL would be the major lipase to respond a “fasting” in *C. elegans*^{5,6}. Therefore, in the present study, we have focused on the regulation of ATGL by PEX5.

Nevertheless, we have tested whether HSL might be regulated by PEX5. As shown in the Reviewer’s only Figure 1a, it seems that HSL also physically interacted with PEX5. In addition, suppression of PEX5 attenuated HSL translocation onto LD fraction during ISO-stimulated lipolysis (Reviewer’s only Figure 1b), implying that HSL seems to be also regulated by PEX5. However, when we compared the contribution of ATGL and HSL to ISO-stimulated lipolysis, suppression of ATGL more potently blocked stimulated lipolysis than that of HSL (Reviewer’s only Figure 1c). Of course, we can’t exclude the possibility that HSL might affect fasting-induced lipolysis in adipocytes, which will be further investigated with PEX5 in future study.

Reviewer’s only Figure 1. The involvement of PEX5 in HSL mediated lipolysis.

a, Co-immunoprecipitation with an anti-MYC antibody and western blotting were conducted with the indicated antibodies. HEK 293 cells were transfected with MYC-PEX5 and GFP-ATGL expression vectors. **b**, Western blot of whole cell extracts (WCE) or LD fractionation of adipocytes transfected with siNC or siPEX5. 30 µg of protein of WCE; 10 µg of protein of LD fraction. **c**, Relative glycerol release from differentiated adipocytes 48 h after siRNA transfection. CON, control; ISO, isoproterenol. n = 3 for each group. *** $P < 0.001$ in two-way ANOVA followed by Turkey’s post-hoc test.

4. Blockade of microtubule polymerization by nocodazole or knockdown of KIFC3 kinesin may not be specific for peroxisome movement but also affect other cellular transport processes. Accordingly, the presented experiments require a negative control to determine the peroxisome-independent effect of nocodazole and KIFC3.

Thank you for this comment. We also agree the idea that the microtubule and motor proteins would be involved in the transport of other subcellular organelles. Nonetheless, the effect of microtubule polymerization on lipolysis is poorly understood. As there are technical difficulties to get rid of peroxisome selectivity, it is very difficult to test peroxisome-independent effects on lipolysis by either nocodazole or KIFC3 knockdown. Thus, we decided to adopt an alternative approach to confirm the

specificity of KIFC3 on peroxisome translocation along the microtubule. For example, to test the specificity of motor protein for peroxisome translocation, we examined the effect of KIF5B, which is important for microtubule dependent movement of other subcellular organelles. As shown in new Supplementary Figure 2b, KIF5B suppression had no effect on peroxisome movement. Also, inhibition of KIF5B did not alter ISO-stimulated lipolysis (new Supplementary Figure 2f and g).

Previously, it has been reported that the peroxisomal protein PEX1 interacts with KIFC3 and acts as a peroxisomal adaptor protein for KIFC3^{7,8}. To investigate PEX1 might affect specific movement of peroxisome to LDs, we examined peroxisome localization with or without PEX1 suppression. As shown in a new Supplementary Figure 2b, PEX1 suppression diminished peroxisome movement to LDs in ISO-treated adipocytes. Furthermore, lipolytic activity was attenuated by PEX1 suppression upon ISO treatment (new Supplementary Figure 2d and e). Together, these findings suggest that KIFC3 would be involved in peroxisome translocation to mediate fasting-induced lipolysis. We described these data in the revised manuscript (pp. 6~7).

5. *KIFC3 kinesin knock down efficiency should be documented by either western blot or qRT-PCR.*

According to this comment, we provided the data for KIFC3 suppression in a new Supplementary Figure 2e as well as others. We described these data in the revised manuscript (p. 7).

6. *The conclusion that peroxisome proliferation in response to WY14643 affects lipolysis may be incorrect because WY not only affects peroxisome abundance but also ATGL transcription. Accordingly the results displayed in Fig. 2g-i are not very informative.*

The points from this comment are well taken. We have investigated the levels of ATGL mRNA upon WY-14643. As the reviewer pointed out, *Atgl* mRNA was increased by WY-14643 (new Supplementary Fig. 2i). However, basal lipolysis was not altered by WY-14643 (new Figure 2i). Also, we found that the contact between peroxisomes and LDs was selectively further increased by WY-14643 in the presence of ISO (Figure 2g, h). Therefore, we believe that WY-14643 would increase *Atgl* mRNA and stimulate contacts between peroxisomes and LDs during ISO-stimulated lipolysis, but not basal lipolysis. We described this issue in the revised manuscript (p. 7 and p. 15).

7. *The description of lipolytic parameters in PEX5-AKO mice is quite preliminary and incomplete. Total adipose mass and the masses of specific adipose depots should be presented. Also, the contribution of ATGL and HSL to FA and glycerol release should be determined using specific enzyme inhibitors. Potential lipid accumulation in other organs such as liver and heart may be indicative of a potential regulatory role of peroxisome for lipolysis in non-adipose tissues! What about LD localization of ATGL and PMP70 in PEX5-AKO mice???*

We appreciate this comment. According to this critique, we have examined whole body weights and tissue weights of epididymal white adipose tissue (eWAT) and inguinal white adipose tissue (iWAT) from WT and PEX5 AKO mice. As shown in new Figure 7a-c, body weight was not altered by PEX5 ablation. During fasting, PEX5 deficient mice showed less body weight reduction than WT. Likewise, the weights of eWAT and iWAT were less decreased in PEX5 AKO mice than WT mice.

In addition, we have also elucidated the contribution of ATGL to PEX5-mediated lipolysis in WAT. As shown in Figure 7j, new Supplementary Figure 5a and b, lipolytic activity was attenuated in eWAT

of PEX5 AKO upon fasting or ISO treatment. Moreover, PEX5 ablated eWAT showed no further inhibitory effect on lipolytic activity when ATGL was pharmacologically inhibited by Atglistatin, implying that PEX5 and ATGL might act in the same pathway.

Upon this comment, we investigated liver weight and histology of PEX5 AKO mice upon fasting. As shown in new Supplementary Figure 5d and e, PEX5 ablation in adipocytes did not significantly alter liver weight and overall morphology of liver upon fasting, compared to WT mice.

In addition, we examined the subcellular localization of peroxisomes in eWAT upon fasting. As shown in the new Supplementary Figure 5c, the interaction between peroxisomes and LDs upon fasting was not affected by PEX5 ablation. We described these data in the revised manuscript (pp. 12~13)

8. Interaction studies between ATGL and PEX5 are preliminary and require further experimental proof by proximity ligation assays or FRET.

Following this comment, we have performed *in situ* proximity ligation assay (PLA). As shown in the new Figure 6d, PLA-positive signals were further increased by FSK treatment. With this PLA assay as well as biochemical and cell biology data (Figure 6) indicate that ATGL would physically interact with PEX5. We described this in the revised manuscript (p. 11)

9. The effect of PEX5 silencing in the absence of CGI-58 on lipolysis seems rather minor. Accordingly, the conclusion that PEX 5 regulates lipolysis independently of CGI-58 requires better experimental support.

To validate and confirm the contribution of PEX5 and CGI-58 to stimulated lipolysis, we have repeated lipolysis analyses. Compared to single knockdown of either PEX5 or CGI-58, co-suppression of PEX5 and CGI-58 group revealed additional inhibitory effects on stimulated lipolysis (Supplementary Figure 6a, lane 3, 4, and 5). Furthermore, colocalization of ATGL and PEX5 at LD surfaces was not affected by CGI-58 suppression (Supplementary Figure 6c). Taken together, these data imply that PEX5 and CGI-58 might contribute to stimulated lipolysis, probably, through a different mode of action. We modified the text to tone down it (p. 14 and p. 18).

10. Fig. 6d lacks an experiment GFP-ATGL+; Myc-PEX5- in the presence or absence of forskolin. Without is the claim that ATGL activity depends on PEX5 is not justified.

Thank you for this comment. According to this comment, we included the data from GFP-ATGL or MYC-PEX5 expression group in the new Figure 6e. We also described this in the revised manuscript (p. 11)

Reviewer #2 (Remarks to the Author):

The authors found that physical interaction between peroxisomes and lipid droplets (LDs) occurs in differentiated 3T-L1 adipocytes upon fasting, which stimulates glycerol release by enhancing

lipolysis. The author also showed that PEX5 interacts with adipose triacylglycerol lipase (ATGL), a rate-limiting enzyme for TG lipolysis, and efficiently colocalizes with it on LDs in the presence of fasting cues. Furthermore, the recruitment of ATGL to LDs was impaired in adipocytes transfected siRNA against PEX5 as well as adipocytes derived from adipose-specific PEX5-knockout mice, resulted in attenuation of fasting-induced lipolysis. From these results, the authors proposed a working model that upon fasting PEX5 recognizes and binds to ATGL, the resultant PEX5-ATGL complex translocates to the contact points (so to speak “contact sites”?) between peroxisomes and LDs, eventually leading to fasting-induced lipolysis. These findings are interesting and important to uncover the molecular mechanism of tightly regulated lipolysis in adipocytes. However, translocation of PEX5-ATGL complex to the contact points between peroxisomes and LDs was not clearly shown. In addition, there are several critical issues that need to be addressed as summarized below.

Major comments

1. The authors showed that physical contacts of peroxisomes with LDs were augmented upon fasting conditions, which was correlated with fasting-induced lipolysis (Figs 1 and 2). However, it is not clear whether physical contacts between peroxisomes and LDs were essential for lipolysis upon fasting or just a response to fasting. Indeed, peroxisome itself appears to be dispensable for fasting-induced lipolysis as assessed by normal lipolysis in the PEX19 (prx-19) RNAi-treated worms (Fig. 4a). Are physical contacts between peroxisomes and LDs required for fasting-induced lipolysis and accumulation of ATGL and PEX5 on LDs? Please explain them with direct evidence including localization of ATGL and PEX5 in PEX19-siRNA-treated adipocytes upon fasting.

We appreciate this comment. In order to avoid inappropriate misunderstandings, we replaced the heatmap data of previous Figure 4a with bar graph data of Oil Red O (ORO) staining through RNAi screening and marked statistical indicators as asterisks by using graphpad prism 7 software. As shown in the new Figure 4a, suppression of *prx5* potentially blocked fasting-induced lipolysis while suppression of *prx-1*, *prx-13*, and *prx-19* also exhibited slightly but statistically significant reduction in ORO intensities.

According to this comment, we have examined whether mammalian PEX19 might be involved in fasting-induced lipolysis in adipocytes. As shown in new Supplementary Figure 7a and b, suppression of PEX19 in adipocytes downregulated ISO-stimulated lipolysis. Furthermore, PEX19 suppression attenuated ISO-induced ATGL translocation onto LDs and inhibited PEX5 localization at LD surface (new Supplementary Figure 7c). We described this in the revised manuscript (p. 17).

As described above [the answer for the comment No.4 of reviewer 1], we have shown that physical contact between peroxisomes and LDs is crucial for fasting-induced lipolysis by treating nocodazole and suppression of KIFC3 and PEX1 in adipocytes (Figure 2 and new Supplementary Figure 2). Together, these data propose that physical contact between peroxisomes and LDs would be important for translocation of ATGL and PEX5 onto LDs.

2. The authors found that translocation of peroxisomes but not mitochondria to LDs upon fasting

and treatment with ISO or forskolin. Although the physical contacts between peroxisomes and LDs were shown by the presence of PMP70, a peroxisomal membrane protein, in the fractions enriched in LDs (Fig. 1i), distribution of peroxisomal matrix and mitochondrial proteins should be shown to strengthen the data in Fig. 1b and Supplemental Fig. 1d.

Thanks for this comment. As the reviewer suggested, we have investigated the subcellular localization of peroxisomal Catalase by ISO treatment. As shown in the new Supplementary Figure 1e, the localization of peroxisomal Catalase at LDs was increased by ISO treatment. In addition, TMRE, mitochondrial marker, revealed that movement of mitochondria toward LDs in adipocytes was not altered by ISO treatment (new Supplementary Figure 1g). We described these data in the revised manuscript (p. 6).

3. The authors claimed that PEX5 escorts ATGL to the contact points between peroxisomes and LDs in the presence of fasting cues. However, the localization of ATGL and PEX5 on the contact points between peroxisome and LDs was not directly shown, though ATGL was largely colocalized with PEX5 on LD upon forskolin treatment (Fig. 6a). This must be demonstrated.

We also agree that this comment is critical to understand the roles of peroxisome and PEX5 in ATGL translocation. Since the immersion oils, required for each laser excitation, need to be finely controlled, there is a technical limitation, causing the resolution to be considerably deteriorated when several signals are monitored by super resolution microscopy at the same time. Thus, it is very difficult to observe simultaneously ATGL, PEX5, peroxisome, and LD in a large focal area. Nonetheless, as shown in the new Supplementary Figure 4d, we were clearly able to observe the colocalization of ATGL and PEX5 on the contact points between peroxisomes and LDs. These new data are included in the revised manuscript (p.10).

4. By suppressing the expression of peroxisomal protein(s) using RNAi, the authors found that suppression of PEX5 (prx-5) but not other PEX (prx) genes attenuated LD hydrolysis upon fasting in C. elegans (Fig. 4a). However, it is expected that several functions of peroxisome be compromised by the knockdown of PEX5 as well as other PEX genes. So that what is the physiological consequence of the formation of contact between peroxisomes and LDs? Is there any specificity in the fatty acids of triacylglycerol during lipolysis upon fasting?

As described above [the answer for the comment No.1 of reviewer 2], we found that other *prx* and PEX genes are also involved in fasting-induced lipolysis (new Figure 4a, new Supplementary Figure 7a, and b). In this study, our data propose that PEX5 could regulate ATGL translocation at the contact points between peroxisomes and LDs, and that other PEX genes, such as PEX3 and PEX19, would be associated with the contacts between peroxisomes and LDs through regulating peroxisome biogenesis. We described these in the revised manuscript (p. 17).

Although further studies are required to elucidate the roles of contacts between peroxisomes and LDs, we have shown that physical contacts between peroxisomes and LDs were required for ATGL translocation, which was escorted by PEX5 upon fasting. Unlike mitochondria, peroxisomes rapidly moved toward LDs and mediated ATGL translocation to initiate lipolysis upon fasting cues (Figure 2, 3 and Supplementary Figure 1). These findings suggest that the crosstalk between peroxisomes and LDs would be an important node for lipolysis initiation upon fasting signals. For long time, the

specificity of peroxisomal and mitochondrial fatty acid oxidation upon fasting has been poorly understood. Recently, it has been reported that the contact between peroxisomes and mitochondria stimulates the transport of acetyl-CoA from peroxisomes to mitochondria⁹. In addition, it has been shown that peroxisomes have carnitine acyltransferase activity, which is required for converting fatty acyl CoAs to acylcarnitines so that they can be transferred to mitochondria for further oxidation¹⁰. Furthermore, various types of fatty acids, including very long chain fatty acids and polyunsaturated fatty acids which are preferentially oxidized by peroxisomes, are released from the triacylglycerol by lipolysis¹⁰⁻¹². Therefore, it is likely that the interaction between peroxisomes and LDs might provide fatty acyl CoA by initiating lipolysis and preferentially oxidizing fatty acids. We described these in the revised manuscript (p. 16)

5. Fig. 6: The authors showed the interaction between ATGL and PEX5 by immunoprecipitation assays. Do they interact directly and how does PEX5 interact with ATGL harboring no typical peroxisome-targeting signal type 1 (PTS1)? And an equally important issue including mechanistic insight of the PEX5-mediated localization of ATGL to the contact points between peroxisomes and LDs should be shown.

As rightly noted by the reviewer, ATGL does not have a conventional C-terminal signal that could serve as the PTS1. Based on publications of Walker et al. claiming that TSC2 and ATM have an internal PTS1, we have investigated whether any such signal might be present in ATGL^{13,14}. Among mouse ATGL amino acid sequences, we found an ARL sequence motif at position 304 (Reviewer's only Figure 2a). Upon mutation, we observed that the interaction between PEX5 and ATGL was impaired, and that increased lipolysis induced by forskolin (FSK) was downregulated (Reviewer's only Figure 2a-c). However, based on personal communications and extensive discussion from the experts in peroxisome biology and co-author Dr. Myriam Baes, we did not further consider the internal ARL sequence as a genuine PTS1 targeting signal, because the latter needs to be free at the C-terminus. As it has been reported that PEX5 is able to translocate target proteins independently of the canonical PTS1¹⁵⁻¹⁸, it needs to be further elucidated how PEX5 can recognize and translocate ATGL upon fasting via a PTS1-independent manner.

To prove which sequence motif of ATGL would be responsible for interacting with PEX5 would require tremendous efforts, which would be an independent future study. Nevertheless, to further substantiate the interaction of PEX5 with ATGL, we performed *in situ* proximity ligation assay (PLA). As shown in the new Figure 6d, PLA-positive signals were increased by FSK treatment. We described this in the revised manuscript (p. 11).

Reviewer's only Figure 2. The reduction of binding between ATGL and PEX5 by mutation of ARL motif of ATGL.

a, Schematic domains of the ATGL (amino acids 486) showing the PEX5-binding sequence (PxBS) and point mutation arginine to glutamine (ATGL RQ mutant). **b**, Co-immunoprecipitation with an anti-MYC antibody and western blottings were conducted with the indicated antibodies. HEK 293 cells were transfected with MYC-PEX5, GFP-ATGL WT, and GFP-ATGL RQ expression vectors. **c**, Concentrations of glycerol released in media by cultured cells transfected with PEX5-WT, ATGL-WT, and GFP-ATGL RQ expression vectors. Cells were pretreated with oleic acid (500 μ M) for 48 h and FSK (25 μ M) for 3 h. $n = 3$ for each group. CON, control; FSK, forskolin. Data represent the mean \pm SD; *** $P < 0.001$ by two-way ANOVA followed by Turkey's post-hoc test.

6. Fig. 6e: Phosphorylation of MycPEX5 was demonstrated by immunoblot using pPKA antibody that detects peptides and proteins containing a phospho-serine/threonine residue with arginine at the -3 position. Is there any putative phosphorylation site(s) in PEX5? Phosphorylated PEX5 upon PKA activation with forskolin and the reduced level of phosphorylated PEX5 upon phosphatase treatment should be demonstrated. Why was the signal corresponding to the PKA-mediated phosphorylated PEX5 in lane 3 higher than that in the case of both expressed together with GFP-ATGL in lane 4?

We have examined putative PKA phosphorylation sites in PEX5 by using GPS3.0 software. There are 10 putative phosphorylation sites in PEX5 protein (Reviewer's only Figure 3). When we investigated the levels of PEX5 phosphorylation, the band intensity of phosphorylated PEX5 was too

weak to distinguish between lane 3 and 4 (previous Figure 6e). Instead, we have repeated new western blot analyses to detect the degrees of PEX5 phosphorylation. As shown in a new Figure 6f, FSK increased PEX5 phosphorylation and enhanced the interaction with ATGL. However, FSK-induced interaction between ATGL and PEX5 was reduced by phosphatase treatment. In addition, PEX5 phosphorylation was not affected by expression with ATGL (New Figure 6f lane 3 and 4).

Position	Code	Kinase	Peptide	Score	Position	Code	Kinase	Peptide	Score
435	S	AGC/PKA	LRDWLRYSPAYAHLV	8.667	281	S	AGC/PKA	VEFERAKSAIESDVD	4.21
263	T	AGC/PKA	EAWVDQFTRPGNKIA	6	589	S	AGC/PKA	ALNMQRKSRGPRGEG	4
51	S	AGC/PKA	ASAAETVSKPLGVGT	5.667	629	S	AGC/PKA	AADARDLSALLAMFG	3.832
149	T	AGC/PKA	QEFIAEVTDPLSVSP	5.333	229	S	AGC/PKA	QIGEGQVSLESAAGS	3.768
155	S	AGC/PKA	VTDPLSVSPARWAE	4.667	435	S	AGC/PKA	LRDWLRYSPAYAHLV	3.642

Reviewer’s only Figure 3. Putative PKA phosphorylation sites of PEX5.

Potential PKA phosphorylation site prediction results obtained using the GPS 3.0 software.

7. Fig. 8: There was no result demonstrating that either ATGL or PEX5 is recruited to the contact points between peroxisomes and LDs. In addition, dephosphorylation of PEX5 that had been recruited on the contact points between peroxisomes and LDs was not shown. Are PEX5 and/or ATGL essential for the formation of contact points between peroxisomes and LDs? These issues are critical to review the authors’ working model.

We appreciate for this comment. As described above [the answer for the comment No.3 of reviewer 2], we demonstrated that ATGL and PEX5 complex was recruited to the contact points between peroxisomes and LDs upon ISO (new Supplementary Figure 4d).

Also, we investigated whether PEX5 and ATGL might affect the interaction between peroxisomes and LDs. As shown in new Supplementary Figure 4e and f, the contact between peroxisomes and LDs was not affected by suppression of PEX5 or ATGL upon ISO. Furthermore, the interaction between peroxisomes and LDs upon fasting was not affected in PEX5 AKO (new Supplementary Figure 5c). Together, these data suggest that the PEX5 would not be involved in the formation of contact between peroxisomes and LDs. Rather, we found that PEX5 would play important roles to translocate ATGL onto the contact points between peroxisomes and LDs. Even though we have shown that PEX5 phosphorylation would be important for the physical interaction between PEX5 and ATGL (Figure 6f), it needs to be further elucidated whether PEX5 dephosphorylation might be important for PEX5 translocation to contact points between peroxisomes and LDs. We described these in the revised manuscript (p. 10 and p. 13).

8. In Figs. 3a and 3b, ATGL-1 in fasting worm and ATGL in ISO-treated adipocytes were localized to peroxisomes in the cytoplasm in addition to the peroxisomes in proximity to LDs. Why?

As shown in new Supplementary Figure 4d and the Reviewer’s only Figure 4 (enlarged 1), colocalized ATGL and PEX5 proteins were detected at the contact points between peroxisomes and LDs. On the other hand, as the reviewer pointed out, ATGL-peroxisome complexes were also

observed in the cytoplasm. We have shown that peroxisomes moved toward LDs in FSK-treated live adipocytes (new Supplementary Figure 1f). Thus, there is a possibility that ATGL could localize to the peroxisome via PEX5 in the cytoplasm, while the peroxisome is migrating to LD upon ISO treatment. To validate this, we examined the localization of ATGL, PEX5, and peroxisome in ISO-treated adipocytes. As shown in the Reviewer's only Figure 4 (enlarged 2), we found that ATGL-PEX5 complexes were detected in peroxisomes, which were present in proximity to LDs upon ISO.

Reviewer's only Figure 4. Localization of ATGL, PEX5, and peroxisome during isoproterenol treatment.

Representative SIM images of ATGL, PEX 5, and PMP70 in ISO (isoproterenol)-treated adipocytes immunostained with endogenous PEX5 (green), ATGL (red), and PMP70 (blue). Scale bars, 1 μ m. Closed arrowhead: colocalization of ATGL, PEX5, and peroxisome at LD surfaces. Open arrowhead: colocalization of ATGL, PEX5, and peroxisome which is migrating to LDs.

9. Furthermore, PEX5 was partly present in LD fraction even in normal condition (Fig. 5g). Given the findings that localization of PEX5 to LDs appears to be more efficient than that of PMP70 (peroxisomes) upon fasting (Fig. 2 and 6a) and PEX5 is specifically involved in fasting-induced lipolysis in *C. elegans* (Fig. 4a-d), regulation of fast-induced lipolysis in adipocytes might be explained as a novel function of PEX5, independent of physical contacts between peroxisomes and LDs. The authors need to clearly distinguish between the roles of peroxisome-LD contacts and PEX5 in fasting-induced lipolysis.

We appreciate this comment. As described above [the answer for the comment No.1 of reviewer 2], suppression of *prx-1*, *prx-13*, and *prx-19* also showed slightly but statistically significant decrease of ORO staining in fasted *C. elegans*.

In order to investigate the roles of peroxisome-LD contacts and PEX5 in fasting-induced lipolysis,

we carefully examined the involvement of mammalian PEX genes in adipocytes. Given that PEX3 and PEX19 are involved in peroxisome biogenesis, it has been reported that suppression of PEX3 and PEX19 causes peroxisome deficiency^{19, 20}, eventually blocking the contact formation between peroxisomes and LDs. As shown in new Supplementary Figure 7a and b, ISO-stimulated lipolysis was attenuated by suppression of PEX3 and PEX19. Furthermore, suppression of KIFC3 and PEX1 inhibited the contacts between peroxisomes and LDs, resulting in the reduction of ISO-stimulated lipolysis (Figure 2d-f and new Supplementary Figure 2b-e). In addition, peroxisome defects caused by PEX19 suppression diminished the recruitment of ATGL and PEX5 onto LD surfaces (Supplementary Figure 7c). Also, we observed that lipolytic activity in adipocytes was attenuated by suppression of PEX14, initial peroxisomal docking site for PEX5 (new Supplementary Figure 7a and b). Together, these data imply that the interaction between peroxisomes and LDs would be required for the translocation of ATGL-PEX5 complex onto LDs. We described these in the revised manuscript (p. 7 and p. 17).

10. Import of peroxisomal matrix proteins in *C. elegans* completely depends on PTS1 due to the lack of PTS2 pathway. So, the roles of PEX5 in the peroxisomal matrix protein import processes between *C. elegans* and mammals are likely distinct. Such issues need to be addressed before explaining and discussing the novel function of PEX5 in fasting-induced lipolysis.

The points are well taken. We also agree that this issue needs to be addressed before explaining the role of PEX5 in fasting-induced lipolysis. Currently, we cannot exclude the possibility that there might be different roles of PEX5 in the process of fasting-induced lipolysis because peroxisomal biology may be different in *C. elegans* and mammals. We described this issue in the “Results” part of the revised manuscript (pp. 9~10).

According to this comment, we have examined whether mammalian PTS2 transport pathway might be associated with fasting-induced lipolysis. As shown in the Reviewer’s only Figure 5, PEX7 suppression had no effect on ISO-stimulated lipolysis, implying that mammalian PTS2 transport pathway might not be involved in stimulated lipolysis.

Reviewer’s only Figure 5. The involvement of PEX7 in ISO-stimulated lipolysis.

a, Relative glycerol release by adipocytes transfected with negative control (NC), PEX7 siRNA (siPEX7) for 48 h. CON, control; ISO, isoproterenol. Cells were treated with ISO (1 μ M) for 1 h. n.s., not statistically significant. **b**, Relative mRNA level of *Pex7* gene in adipocytes after siRNA transfection (48 h). Data represent the mean \pm SD; *** P < 0.001 vs. siNC-CON in two-way ANOVA followed by Turkey’s post-hoc test.

Although further studies are required to fully understand detail mechanisms for PRX-5/PEX5-mediated lipolysis in *C. elegans* and mammals, this study reveals that the regulation of ATGL-1/ATGL would be mediated by PRX-5/PEX5 upon fasting.

Minor comments

1. Figs. 1i and 5g: Relative amounts of LD fractions to whole cell extracts (WCE) should be shown. Distribution of PMP70 needs to be shown in Fig. 5g. This is critical to validate the efficiency of physical contacts between peroxisomes and LDs.

We provided the information of protein amounts of LD fractions and WCE in the legends of Figure 1 and 5 (p. 33 and pp. 36~37). Following this comment, we included western blot data for PMP70 in Figure 5g.

2. Fig. 3a: Why were most of ATGL-1 co-localized with DS-RED-PTS1 even at the fasting period 0 hour?

We provided quantitation data in a new Figure 3b. The basal level of colocalization between ATGL-1 and DS-RED-PTS1 was about 27% of total ATGL-1 protein. It seems that ATGL-1/ATGL was colocalized with peroxisome under basal status (new Figure 3b, d, and e). More importantly, colocalization between peroxisomes and ATGL was further elevated by fasting signals.

3. Fig. 5, a-c: The levels of relative glycerol release by the ISO-treated adipocytes that were transfected with negative control siRNA (siNC, solid bars) are much higher in 5a than in 5b and 5c despite the same experimental condition. Why is that so?

The levels of glycerol release seem to be variable upon degree of adipocyte differentiation. For previous data preparation, we used different cohorts of different adipocytes (previous Figure 5a, b, and c). According to this comment, we repeated glycerol release assays in the same cohort of differentiated adipocytes to obtain new Figure 5a, b, and c.

4. Fig. 5c: How much were the protein level of ACOX1 and/or peroxisomal fatty acid oxidation level suppressed in this experimental condition?

According to this comment, the levels of *Acox1* mRNA and protein were provided in the new Supplementary Figure 4b.

5. Fig. 5 h: Statistical analysis is required.

We included statistical analysis for relative levels of ATGL and described the information of statistical analysis in the legend of Figure 5. We quantified the band intensity of ATGL protein from 4 independent experiments. Suppression of PEX5 diminished ATGL translocation into the LD fraction during ISO-stimulated lipolysis (new Figure 5g and h).

6. Fig. 6, b, c, and e: Size markers should be indicated.

We included size markers in new Figure 6b, c, and f.

7. Fig. 6d: Expression of only GFP-ATGL should be included as a control to verify the effect of PEX5 in lipolysis.

As described above [the answer for the comment No.10 of reviewer 1], we included the data from either GFP-ATGL or MYC-PEX5 in a new Figure 6e. We described this in the revised manuscript (p. 11)

8. Supplementary Figure 1d: Mitochondrial signal was too weak in the presence of ISO.

As described above [the answer for the comment No.2 of reviewer 2], we replaced a new Supplementary Figure 1g with live Representative SIM images of adipocytes stained with BODIPY (LD marker) and TMRE (mitochondria marker).

9. Supplementary Figure 5h: Statistical analysis is required.

We included statistical analysis in a new Supplementary Figure 6h (previous Supplementary Figure 5h).

10. Several recent papers reported on the relationship between peroxisomes and LDs (Schrul and Kopito, *Nat. Cell Biol.* 18: 740-751 (2016), Wang et al., *Nat. Commun.* 9: 2939 (2018), Joshi et al., *Nat. Commun.* 9: 2940 (2018)). They should be properly cited and discussed.

Thank you for this suggestion. We added the above papers into the “Introduction” part of the revised manuscript (p. 4).

Reviewer #3 (Remarks to the Author):

This is an excellent study that enriches the presently limited but growing body of knowledge around the peroxisome's role in lipid metabolism in adipose tissues. Here, Kong et al investigate the mechanism by which peroxisomes and lipid droplets (LD) tether to each other in adipocytes during fasting. Previous studies have shown that peroxisomes may have a role in regulating LD in adipose tissue. Using multiple systems including culture cells, C. elegans and mouse models, Kong et al show that the peroxisomal matrix import protein receptor, PEX5, is not only required to form the peroxisome-LD tether but also recruit ATGL to LD during fasting.

The work begins with the demonstration of an increase in peroxisome-LD interaction during fasting in C. elegans, mouse epididymal adipose tissue and 3T3-L1 in a microtubule and KIFC3 dependent fashion. Using the same systems, they also show that PEX5, a cytosolic peroxisomal matrix protein receptor, is required to recruit ATGL to LD assumingly to the site where the peroxisome-LD tether is formed; this tether is required for the lipolysis during fasting in adipose tissue. Finally, they present that in adipose tissue from PEX5 KO mice, ATGL recruitment is defective and show reduced lipolysis. While defect in adipose tissue development has previously been demonstrated in these aP2-PEX5 KO mice, the finding here suggests a possible mechanism by which peroxisomes regulate LD in adipose tissues. However, in order to be considered for publication in Nature Comm, the authors need to consider the following points about this study and make

appropriate experimental and presentation-related amendments.

*1) It is not clear where and how ATGL is localized with peroxisomes. The authors make a convincing argument for PEX5-ATGL interaction. However, how is PEX5 recruiting ATGL to LD? Does it required Peroxisomes? If peroxisomes as a whole is required for ATGL recruitment to LD, is PEX5 recruiting ATGL into the matrix or just to the surface? In Figure 3a, ATGL appears to colocalize with DS-RED-PTS even before fasting, suggesting that ATGL is localized to peroxisomes prior to being targeted to LD. Since PEX5 is the matrix protein cytosolic receptor, one would postulate that ATGL would be imported into peroxisomes. If this is the case, how would ATGL act on the LD? However, if PEX5 recruits ATGL to the surface of peroxisomes and LD, then there may be other other cellular sequences such as peroxisome degradation. It has previously shown that any aborted import that results in PEX5 accumulation on the peroxisomal membrane has been shown to induce peroxisome degradation. In other words, if ATGL recruitment requires PEX5 on the peroxisomal membrane as the author states, then this may induce pexophagy due to the accumulation of membrane bound PEX5. It may be difficult to imagine that ATGL would be in the matrix of the peroxisomes as it postulated function is to activation of lipolysis of, however, such a mechanism may be possible if peroxisomes protrudes into LD as shown for *S.cerevisiae*. This is an important point as ATGL localization inside peroxisomes suggests a novel mechanism for lipolysis in LD. Similarly, if PEX5 recruits ATGL to the peroxisomal surface or just to the peroxisomal matrix import pathway, it suggests a novel import pathway that is mediated by a peroxisomal protein. To address the sub-cellular localization, the authors should attempt the protease protection assay in intact cells using agents that semi-permeabilize the plasma membrane such as digitonin.*

We really appreciate this valuable comment and suggestion. According to the reviewer's critique, we have performed protease protection assays treated with proteinase K in the absence or presence of Triton X-100. As shown in a new Figure 3e, the levels of ATGL protein were increased in the peroxisome fraction of ISO-treated adipocytes. Similar to the peroxisomal membrane protein PMP70, but not the peroxisomal matrix protein Catalase, ATGL protein was degraded by proteinase K in the absence and presence of Triton X-100. Furthermore, when we examined the subcellular location of peroxisomal Catalase and ATGL with super resolution microscopy (SIM), Catalase was detected inside the peroxisome whereas ATGL was observed at the surface of peroxisomes (new Supplementary Figure 3a and b). These data suggest that ATGL would be recruited to peroxisomal membrane area, probably, not in peroxisomal matrix upon ISO treatment. We described these in the revised manuscript (p. 8)

To investigate whether autophagy might be involved in stimulated lipolysis, we decided to examine autophagy induction at the time point when ATGL was recruited by PEX5. As shown in the Reviewer's only Figure 6, autophagy was not induced by FSK treatment for 1 hour. Although it needs to be elucidated whether peroxisome specific autophagy (pexophagy) might be associated with ATGL translocation by PEX5 in future study, it seems that overall autophagy might not be involved in early lipolysis in which ATGL was translocated to LD surface areas by PEX5.

Reviewer's only Figure 6. Autophagy alteration upon FSK

Mouse primary adipocytes were treated with or without FSK (10 μ M) for 1 h. After isolating whole cell lysates, western blotting analyses were performed with the indicated antibodies. Adipocytes were treated with or without rapamycin (Rapa, 150 nM) or chloroquine (CQ, 20 μ M) for 5 h.

As described above [the answer for the comment No.9 of Reviewer 2], our data suggest that peroxisomes would be important for ATGL translocation onto LDs, which is escorted by PEX5 upon fasting. In adipocytes, suppression of PEX3 or PEX19 downregulated ISO-stimulated lipolysis (new Supplementary Figure 7a and b). Also, peroxisome defect caused by PEX19 suppression inhibited the recruitment of ATGL and PEX5 onto LD surfaces (new Supplementary Figure 7c). Furthermore, in adipocytes, ISO-stimulated lipolysis was attenuated by suppression of PEX14, an initial peroxisomal docking site for PEX5 (new Supplementary Figure 7a, b).

2) Similarly, it is not clear whether peroxisomes themselves are required for PEX5 to recruit ATGL to the site of Peroxisome-LD contact. It may be that PEX5 alone may be necessary. Based on the RNAi screen for peroxisome proteins required for lipolysis in *C. elegans*, it appears that only PEX5 and not its membrane receptors, PEX14 and PEX13, is needed. Does this mean that PEX5 alone is required? This question must be addressed within the scope of this study to clarify the role of PEX5 and/or peroxisomes in LD regulation. To test this, the authors should use peroxisome-deficient cells to determine whether PEX5 alone is necessary for the recruitment of ATGL to LD.

As described above [the answer for comments No.1 and 9 of reviewer 2], we replaced the heatmap data (previous Figure 4a) with quantification data of Oil Red O (ORO) staining via RNAi screening (new Figure 4a). Suppression of *prx-1*, *prx-13*, and *prx-19* also showed slightly but statistically significant decrease of ORO staining in fasted *C. elegans*. In addition, as described above [the answer for comments No.1 of reviewer 3], peroxisome defect caused by PEX19 suppression repressed the recruitment of ATGL and PEX5 onto LD surfaces (new Supplementary Figure 7c).

To examine whether peroxisomes might be required for ATGL recruitment at peroxisome-LD contact points by PEX5, we carefully investigated the roles of mammalian PEX genes in lipolysis. As shown in new Supplementary Figure 7a and b, lipolytic activity in adipocytes was attenuated by suppression of PEX14. Furthermore, co-suppression of PEX5 and PEX14 showed no further inhibitory effect on lipolytic activity, with a similar knockdown efficiency of PEX5 and PEX14 (Reviewer's only Figure 7). These data imply that membrane receptor PEX14 might be required for PEX5 mediated lipolysis.

Reviewer's only Figure 7. The requirement of peroxisomal genes in the regulation of fasting-induced lipolysis.

a, Relative glycerol release by adipocytes transfected with negative control (NC), PEX5 siRNA (siPEX5), and PEX14 siRNA (siPEX14) for 48 h. CON, control; ISO, isoproterenol. Cells were treated with ISO (1 μ M) for 1 h. Data represent the mean \pm SD; *** P < 0.001 vs. siNC-ISO in two-way ANOVA followed by Turkey's post-hoc test. **b**, Relative mRNA levels of *Pex5* and *Pex14* genes in adipocytes after siRNA transfection (48 h). Data represent the mean \pm SD; *** P < 0.001 vs. siNC-CON in two-way ANOVA followed by Turkey's post-hoc test.

3) Using the quantification of total PMP70 fluorescent intensity from a confocal images, the authors suggest that PMP70 does not increase; hence peroxisomes do not increase. However, in Figures 1c and 1d, the peroxisomes during fasting look sizably larger then in fed conditions, and in Fig. 1g, there appears to be more PMP70 in the ISO-treated WCL. Using fluorescent intensity from immunofluorescent images can be problematic and inaccurate. Is it possible that there are more as well as larger peroxisomes in the adipose during fasting? Given that β -adrenergic receptor induces PPAR-alpha, a nuclear receptor that induces peroxisome proliferation, could it be possible that ISO treatment is inducing an increase in peroxisome size and numbers, which causes an increase in peroxisome-LD interaction?

The points are well taken. To address this issue, we carefully analyzed the sizes of peroxisomes in *C. elegans* and adipocytes. We measured the areas of PMP70 fluorescence positive peroxisomes and classified peroxisomes by size using LAS X (Leica) software. As shown in new Supplementary Figure 1c and d, overall sizes of peroxisomes were not significantly altered upon fasting. In order to avoid any confusion or misunderstanding, we replaced the previous Figure 1c. In a new Figure 1c, we observed that peroxisomes moved to LDs upon fasting.

According to this comment, we have examined the mRNA levels of PPAR α target genes. The levels of PPAR α target genes were not largely changed by ISO treatment for 1 hour. These data propose that peroxisomal size might not be altered by ISO. We described this issue in the revised manuscript (p. 5)

4) It is very difficult to assess some of the figures as information is either hidden or missing. For example, there is no explanation for the way the fluorescent intensity of PMP70 was quantified. Similarly, there is no information about the colocalization quantification. Such information is

critical for the reader to assess the data.

We appreciate this comment. In the revised manuscript, we tried to include detail description of how to quantify PMP70 intensity and colocalization ratio in the Methods section (pp. 22~23).

5) Furthermore, the methods section in general needs improvement. The brevity of the method section made it difficult for this reviewer to fully understand the experiments and some assumptions that have been made. This is especially important as the authors have used multiple model systems and greater clarity is required for readers to understand each of the different systems. For example, what is relative CARS intensity? Another example is the settings for microscopy. While the authors have defined the abbreviations in the main text, providing full descriptions in the methods section would be useful. Greater details are also needed in the figure legend.

Again, thank you for this comment. We provided more information for the relative CARS intensity and how to quantify lipid contents in *C. elegans* using CARS microscopy in the Methods section. In addition, we added detail information for microscopic analyses in the Methods section (pp. 22~23).

6) Does the translocation of peroxisomes to LD require ATGL? This question should be easy to address and would help test whether ATGL and PEX5 form the peroxisome-LD tether.

As described above [the answer for the comment No.7 of reviewer 2], we investigated whether PEX5 and ATGL might be involved in the formation of contact points between peroxisomes and LDs. As shown in new Supplementary Figure 4e and f, the degree of interaction between peroxisomes and LDs were not altered by suppression of PEX5 or ATGL upon ISO. We described these results in the revised manuscript (p. 10).

7) Why is PEX5 in the LD fraction in the non-ISO condition (Fig. 5g) while there is no PMP70 (Fig 1i)? Is it possible that PEX5 targets to LD independently of peroxisomes? If this is the case, it may explain why the depletion of PEX14 does not prevent a reduction of LD as seen with PEX5 knockdown.

As shown in several images of new Figure 1-3, we observed that some peroxisomes appeared to contact with LDs under basal status. In a previous Figure 5g (in the submission version), the protein amounts of LD fraction were too little to detect PMP70 in basal status. In order to detect basal level of PMP70 in the LD fraction, we had to enrich protein levels in the LD fraction, and we replaced the new western blot with a Figure 5g. As shown in a new Figure 5g, peroxisomes were present in the LD fraction under basal status. To date, we have no clue nor evidence that PEX5 might localize to LD independently of peroxisome.

8) Fig 6d: what happens to glycerol release levels when expressing ATGL or PEX5 alone? Is the effect (lipolysis stimulation) additive or dependent?

As described above [the answer for the comment No.10 of reviewer 1 and the minor comment No. 7 of reviewer 2], we included the data for GFP-ATGL or MYC-PEX5 expression group in a new Figure 6e. In the presence of FSK, stimulated lipolysis was increased by overexpression of ATGL or PEX5 alone. When ATGL was coexpressed with PEX5, glycerol release was further elevated by FSK,

implying that the physical interaction between ATGL and PEX5 would be important to mediate stimulated lipolysis. We described this in the revised manuscript (p. 11).

Minor:

Fig. 1f: Recommend showing the enlarged images with both PLIN1 and PMP70 (instead of PMP70 alone) so that readers can easily see the colocalization between peroxisomes and LDs.

We rearranged a new Figure 1f including enlarged images with PLIN1 and PMP70.

- Fig. 3: Quantification (colocalization) is required for all images.

We included the quantitation data in new Figure 1, 2, 3, 5, 7, Supplementary Figure 1 and 4.

- Fig. 5e: Define how the measurement works in the figure legend and/or method. What does “Recruitment of ATGL1 %” mean here?

We appreciate this comment. ATGL recruitment onto LD surface was calculated by the number of ATGL signals in close proximity ($<0.2 \mu\text{m}$) within LD surface divided by total number of ATGL signals using imageJ software. We provided detail description of how to quantify the recruitment of ATGL in the Methods section (p. 23).

- Fig. 5h: Need to show the number of trials performed and statistical analysis.

We included statistical analysis of relative levels of ATGL and described the number of trials (n=4) in the legend of Figure 5.

- Fig. 6a: Visualization of this figure would be greatly aided with an enlarged region

According to this suggestion, we added enlarged images in a new Figure 6a.

- It is difficult to observe the movement of peroxisomes in the time lapse imaging in Fig. S1c, even in the movie. Perhaps showing the separate channels along with the merge for each time frame would make it easier to view the movement of the peroxisomes towards the LD. The authors should also consider showing divided images for the movies, but with a slower time rate than currently used.

Following this comment, we inserted mCHERRY-PTS channel along with merged images in Supplementary Figure 1f and included divided images for movies with a slower time rate in Supplementary Video 1, 2, and 3.

- Fig. 1b: What does the y-axis “colocalization ratio %” mean? Which parameter is being measured here, intensity, number or area? The authors need to explain it in the figure legend. Also define “white arrowhead”. The authors need to explain in detail how the degree of colocalization was measured/calculated for ALL the figures. I have no idea how their “colocalization ratio” was determined.

We appreciate this comment. Colocalization ratio was quantified as the intensity of peroxisomes colocalized with LDs divided by total intensity of peroxisomes using imageJ, Leica (LAS X), Zen

(Zeiss), and Softworx (GE Healthcare Life Sciences) software. We included detail description of how to quantify colocalization ratio in the Methods section (pp. 22~23). We also added information of arrowhead in legends of Figures 1, 2, 3 and 6.

References

1. Lee, J.H. *et al.* Protein Kinase A Subunit Balance Regulates Lipid Metabolism in *Caenorhabditis elegans* and Mammalian Adipocytes. *J Biol Chem* **291**, 20315-20328 (2016).
2. Bezaire, V. *et al.* Contribution of adipose triglyceride lipase and hormone-sensitive lipase to lipolysis in hMADS adipocytes. *J Biol Chem* **284**, 18282-18291 (2009).
3. Haemmerle, G. *et al.* Defective lipolysis and altered energy metabolism in mice lacking adipose triglyceride lipase. *Science* **312**, 734-737 (2006).
4. Osuga, J. *et al.* Targeted disruption of hormone-sensitive lipase results in male sterility and adipocyte hypertrophy, but not in obesity. *Proc Natl Acad Sci U S A* **97**, 787-792 (2000).
5. Lee, J.H. *et al.* Lipid droplet protein LID-1 mediates ATGL-1-dependent lipolysis during fasting in *Caenorhabditis elegans*. *Mol Cell Biol* **34**, 4165-4176 (2014).
6. Zhang, S.O. *et al.* Genetic and dietary regulation of lipid droplet expansion in *Caenorhabditis elegans*. *Proc Natl Acad Sci U S A* **107**, 4640-4645 (2010).
7. Dietrich, D., Seiler, F., Essmann, F. & Dodt, G. Identification of the kinesin KifC3 as a new player for positioning of peroxisomes and other organelles in mammalian cells. *Biochim Biophys Acta* **1833**, 3013-3024 (2013).
8. Nguyen, T., Bjorkman, J., Paton, B.C. & Crane, D.I. Failure of microtubule-mediated peroxisome division and trafficking in disorders with reduced peroxisome abundance. *J Cell Sci* **119**, 636-645 (2006).
9. Shai, N. *et al.* Systematic mapping of contact sites reveals tethers and a function for the peroxisome-mitochondria contact. *Nat Commun* **9**, 1761 (2018).
10. Lodhi, I.J. & Semenkovich, C.F. Peroxisomes: a nexus for lipid metabolism and cellular signaling. *Cell Metab* **19**, 380-392 (2014).
11. Crockett, E.L. & Sidell, B.D. Peroxisomal beta-oxidation is a significant pathway for catabolism of fatty acids in a marine teleost. *Am J Physiol* **264**, R1004-1009 (1993).
12. Hiltunen, J.K., Filppula, S.A., Hayrinen, H.M., Koivuranta, K.T. & Hakkola, E.H. Peroxisomal beta-oxidation of polyunsaturated fatty acids. *Biochimie* **75**, 175-182 (1993).
13. Zhang, J. *et al.* A tuberous sclerosis complex signalling node at the peroxisome regulates mTORC1 and autophagy in response to ROS. *Nat Cell Biol* **15**, 1186-1196 (2013).
14. Zhang, J. *et al.* ATM functions at the peroxisome to induce pexophagy in response to ROS. *Nat Cell Biol* **17**, 1259-1269 (2015).
15. van der Klei, I.J. & Veenhuis, M. PTS1-independent sorting of peroxisomal matrix proteins by Pex5p. *Biochim Biophys Acta* **1763**, 1794-1800 (2006).
16. Klein, A.T., van den Berg, M., Bottger, G., Tabak, H.F. & Distel, B. *Saccharomyces cerevisiae* acyl-CoA oxidase follows a novel, non-PTS1, import pathway into peroxisomes that is dependent on Pex5p. *J Biol Chem* **277**, 25011-25019 (2002).
17. Elgersma, Y., van Roermund, C.W., Wanders, R.J. & Tabak, H.F. Peroxisomal and

- mitochondrial carnitine acetyltransferases of *Saccharomyces cerevisiae* are encoded by a single gene. *EMBO J* **14**, 3472-3479 (1995).
18. Gunkel, K., van Dijk, R., Veenhuis, M. & van der Klei, I.J. Routing of *Hansenula polymorpha* alcohol oxidase: an alternative peroxisomal protein-sorting machinery. *Mol Biol Cell* **15**, 1347-1355 (2004).
 19. Subramani, S. Components involved in peroxisome import, biogenesis, proliferation, turnover, and movement. *Physiol Rev* **78**, 171-188 (1998).
 20. Hohfeld, J., Veenhuis, M. & Kunau, W.H. PAS3, a *Saccharomyces cerevisiae* gene encoding a peroxisomal integral membrane protein essential for peroxisome biogenesis. *J Cell Biol* **114**, 1167-1178 (1991).

Reviewers' comments:

Reviewer #1 (Remarks to the Author):

The authors addressed all points of criticism and suggestions. I have no further comments.

Reviewer #2 (Remarks to the Author):

Remarks to the Authors:

In this revised manuscript, the authors improved the manuscript by adding new data and revising the Figures. However, the reviewer #2 still has several concerns in the revised manuscript and the authors' replies with regard to the data attempting to support the authors' working model.

Reviewer's comments to the Authors' replies are as follows:

Major points:

1) Authors' reply to the major comment 5 of reviewer #2:

Direct interaction between ATGL and PEX5 is not provided. Positive signals in proximity ligation assay (PLA) are supposed to be observed at contact sites between LDs and peroxisomes in New Fig. 6d. However, the signals are distributed in the cytoplasm and appear to be located on peroxisomes. Did the authors examine PLA assay in HEK293 cells without enhancing LD formation? The authors should clearly describe the experimental condition and precisely explain the results including the issue where ATGL-PEX5 interaction indicated by PLA signal is elevated.

"Reviewer's only Figure 2" appears to include important information that PEX5 can recognize ATGL in a canonical PTS1-independent manner and the interaction between ATGL and PEX5 is required for induction of increased lipolysis by FSK. These would might explain the molecular mechanism underlying the PEX5-mediated ATGL translocation to the peroxisome-LD contact points. However, this issue remains largely unclear even in the revised manuscript. Therefore, the reviewer #2 suggest the authors that "Reviewer's only Figure 2" is to be included in main body and discussed along with the proposed working model (also see comment 2)).

2) Authors' reply to major comments 6 and 7 of reviewer #2:

As the authors reply, newly added data (New Supplementary Figs. 4d-f and 5c) suggest that both ATGL and PEX5 are recruited to the contact points between peroxisomes and LDs, but not essential for their formation. On the other hands, the authors' working model (Fig. 8) evidently illustrates that upon fasting PEX5 is phosphorylated by PKA and forms a complex with ATGL in the cytosol. And the PEX5-ATGL complex is then translocated to the contact points between peroxisomes and LDs, where PEX5 is dephosphorylated. This schema seems to be based on the data of New Fig. 5f showing that FSK induced the elevation of PEX5 phosphorylation and PEX5-ATGL interaction by immunoprecipitation using whole cell lysates. However, the authors provide neither description for phosphorylation-dependent regulation of PEX5-ATGL complex in the working model nor any convincing data demonstrating the dephosphorylation of PEX5 that had been recruited to the contact sites between peroxisomes and LDs. Moreover, the recovery of substantial amount of ATGL in crude peroxisome fraction prepared from cells treated with ISO using sedimentation method (New Fig. 3e) seems to be inconsistent with the working model. The data in this revised Figures do not support the working model. More convincing results are required. Or alternatively, a more appropriate working model based on the experimental data and its explanation are needed.

Minor points:

1. What is TMRE in New Supplementary Fig. 1g?

2. New Supplementary Fig. 2h: Is intensity of PMP70 statistically significant between cells treated with vehicle and WY-14643?

3. Two antibodies each for PMP70 and PEX5 are listed without describing any explanation on page 21, lines 469-471.

Reviewer #3 (Remarks to the Author):

In this revised manuscript, Kong et al provides both new experiments and substantial textual revisions, which address all of my concerns in their first submission. One major concern from the first submission was whether peroxisomes themselves were required for fasting-induced lipolysis and not just PEX5. The authors have addressed this by showing that knockdown of *prx1*, *prx13* and *prx19* in *C.elegans* prevented lipid droplet reduction. This result suggests that the loss of peroxisomes reduces lipolysis during fasting in *C. elegans* as all three of these genes are peroxisome biogenesis factors. Therefore, the assumption here is that the loss of peroxisomes in mammalian cells may also result in reduction in lipolysis during fasting. However, in the revised manuscript, they suggest that the decrease in lipolysis and lack of peroxisome recruitment to lipid droplet in PEX1 knockdown in adipose cells was due to the inability of peroxisomes to bind to the kinesin protein, KifC, which is involved in moving peroxisomes along the microtubule. There are several problems with this interpretation. First, while PEX1 has been shown to bind to KifC3 (Dietrich et al BBA (2013) 1833:2013), it was never shown to be an adaptor for KifC3 in peroxisome mobility. Second, PEX1 has been shown to be required in both *S. cerevisiae* and in cultured mammalian cells to prevent peroxisome degradation by autophagy (Nuttall et al Autophagy (2014) 10:835; Law et al Autophagy (2017) 13:868). Therefore, the lack of or decreased recruitment to lipid droplets may likely be due to the decreased number of peroxisomes. Although the authors show that PMP70 fluorescent intensity is not reduced in PEX1 knockdown in adipose cells, it is possible that there are decreased number of peroxisomes in these cells. This is because the knockdown of PEX1 does not prevent PMPs from targeting to peroxisomes and given that most PMPs have half-lives of less than 16 hours, it is not surprising that PMP70 fluorescents have not decreased within the duration of the RNAi knockdown assay. Furthermore, as pointed out in the previous review, fluorescent intensity is not a good measure of protein levels, especially when using confocal images. To demonstrate that PEX1 knockdown is NOT causing a decrease in peroxisome numbers, the authors should consider quantifying the number of peroxisomes in their adipose cells. Alternatively, the authors should consider removing their interpretation of the PEX1 knockdown data as a supporting evidence to the requirement of KifC3 for peroxisome recruitment to lipid droplet. It is the opinion of this reviewer that the KifB5 data is a sufficient negative control for the KifC3.

Response to the Reviewers' Comments

MS ID#: NCOMMS-18-6227380A

MS TITLE: Spatiotemporal contact between peroxisomes and lipid droplets regulates fasting-induced lipolysis via PEX5

Reviewers' comments:

Reviewer #1 (Remarks to the Author):

The authors addressed all points of criticism and suggestions. I have no further comments.

Reviewer #2 (Remarks to the Author):

Remarks to the Authors:

In this revised manuscript, the authors improved the manuscript by adding new data and revising the Figures. However, the reviewer #2 still has several concerns in the revised manuscript and the authors' replies with regard to the data attempting to support the authors' working model. Reviewer's comments to the Authors' replies are as follows:

Major points:

1) Authors' reply to the major comment 5 of reviewer #2:

Direct interaction between ATGL and PEX5 is not provided. Positive signals in proximity ligation assay (PLA) are supposed to be observed at contact sites between LDs and peroxisomes in New Fig. 6d. However, the signals are distributed in the cytoplasm and appear to be located on peroxisomes. Did the authors examine PLA assay in HEK293 cells without enhancing LD formation? The authors should clearly describe the experimental condition and precisely explain the results including the issue where ATGL-PEX5 interaction indicated by PLA signal is elevated.

"Reviewer's only Figure 2" appears to include important information that PEX5 can recognize ATGL in a canonical PTS1-independent manner and the interaction between ATGL and PEX5 is required for induction of increased lipolysis by FSK. These would might explain the molecular mechanism underlying the PEX5-mediated ATGL translocation to the peroxisome-LD contact points. However, this issue remains largely unclear even in the revised manuscript. Therefore, the reviewer #2 suggest the authors that "Reviewer's only Figure 2" is to be included in main body and discussed along with the proposed working model (also see comment 2)).

According to this comment, we provided detail descriptions for experimental conditions and obtained data in the revised manuscript and Figure legends (pp. 11~12 and p. 37). In HEK293 cells, we noticed the physical interaction between PEX5 and ATGL without lipid challenge. Unlike 3T3-L1 adipocytes, it is well known that LDs of HEK293 cells are very small and rarely present. Also, the PLA assay revealed that the association between PEX5 and ATGL was primarily observed in the cytosol. Thus, both co-immunoprecipitation analysis and PLA assay clearly indicated the physical interaction between PEX5 and ATGL. Further, to investigate the biological significance that the physical interaction between PEX5 and ATGL on LD surfaces would affect lipolytic activity, the level of glycerol release was examined in the presence of lipid challenge. As shown in Figure 6e, stimulated lipolysis was significantly increased by co-overexpression of ATGL and PEX5 in the presence of FSK. Moreover, microscopic analyses also showed that PEX5 and ATGL were colocalized at LD surfaces in adipocytes (Figure 6a and Supplementary Figure 4d).

The reviewer asked to include the "Reviewer's only Figure 2" in the main body of the manuscript. We believe that the essential information of the direct interaction between ATGL and PEX5 has been already shown in the co-immunoprecipitation experiments (Figure 6). In the previous "Reviewer's

only Figure 2”, ATGL was mutated to AQL from an internal ARL sequence (ATGL-RQ mutant). This experiment was performed based on the publication of an internal PTS1 motif in the TSC protein¹. Although we showed that ATGL-RQ mutant was impaired in the binding of PEX5 (previous Reviewer’s only Figure 2b), it has been controversial whether an internal ARL sequence would act as a noncanonical PTS1 targeting signal. According to structural studies, the carboxyl group of the PTS1 sequence is essential for the canonical binding of PTS1 with the TPR motifs of PEX5². After extensive discussion with many experts in peroxisome biology and our co-author Dr. Myriam Baes, it is scientifically not justified that an internal ARL sequence in ATGL is considered as a PTS1 targeting signal. This point has been carefully described in the answer for the 1st reply letter comment No.5 of reviewer 2. On the other hand, it has been reported that PEX5 is able to recognize target proteins independently of the canonical C-terminal PTS1³⁻⁶. Since several peroxisomal proteins also do not have a classical PTS1 nor a PTS2 sequence, it remains unresolved how they translocate into peroxisomes. In addition, we assume that the conformation of the ATGL-RQ mutant might be altered in such a way that the ATGL-PEX5 interaction would be impaired. To test and prove which sequence motif of ATGL would be responsible for interacting with PEX5 requires large efforts, which would be an independent future study. After this study, we will investigate the mechanisms by which PEX5 can recognize and translocate ATGL upon fasting independently of a canonical PTS1. So far, we have not yet obtained enough data to reveal the molecular mechanism underlying the PEX5-ATGL interaction. Therefore, we would prefer not to include the previous “reviewer’s only Figure 2” in the main data.

2) Authors’ reply to major comments 6 and 7 of reviewer #2:

As the authors reply, newly added data (New Supplementary Figs. 4d-f and 5c) suggest that both ATGL and PEX5 are recruited to the contact points between peroxisomes and LDs, but not essential for their formation. On the other hands, the authors’ working model (Fig. 8) evidently illustrates that upon fasting PEX5 is phosphorylated by PKA and forms a complex with ATGL in the cytosol. And the PEX5-ATGL complex is then translocated to the contact points between peroxisomes and LDs, where PEX5 is dephosphorylated. This schema seems to be based on the data of New Fig. 5f showing that FSK induced the elevation of PEX5 phosphorylation and PEX5-ATGL interaction by immunoprecipitation using whole cell lysates. However, the authors provide neither description for phosphorylation-dependent regulation of PEX5-ATGL complex in the working model nor any convincing data demonstrating the dephosphorylation of PEX5 that had been recruited to the contact sites between peroxisomes and LDs. Moreover, the recovery of substantial amount of ATGL in crude peroxisome fraction prepared from cells treated with ISO using sedimentation method (New Fig. 3e) seems to be inconsistent with the working model. The data in this revised Figures do not support the working model. More convincing results are required. Or alternatively, a more appropriate working model based on the experimental data and its explanation are needed.

Thank you for this comment and the points are well taken. Following this comment, we modified the working model and included a more detail description of PEX5-ATGL interaction and translocation to the contact points between peroxisomes and LDs in the legend of the revised manuscript (pp. 38~39). As described in the answer for the 1st reply letter comment No. 7 of reviewer 2, we have shown that PEX5 phosphorylation would be important for the physical interaction between PEX5 and ATGL (Figure 6f). However, we do not have direct evidence that phosphorylation of PEX5 is prerequisite for ATGL translocation to the contact points between peroxisomes and LDs, which will be elucidated in a future study. Thus, we added a question mark in the working model (c). We also described this issue in the legend of the revised manuscript.

As shown in Figure 1i, the interaction between LDs and peroxisomes were increased by ISO. Also, we have shown that the levels of ATGL and PEX5 complex were increased in LDs and peroxisomes by ISO. For example, in the LD fraction, the levels of PEX5 and ATGL were significantly increased by ISO (Figure 5g). In addition, the levels of ATGL protein were increased in the peroxisome fraction of ISO-treated adipocytes even though the levels of ATGL protein from whole cell extracts (WCE) were not largely altered by ISO (Figure 3e). As shown in Supplementary Figure 4d, we also observed the colocalization of ATGL and PEX5 on the contact points between peroxisomes and LDs. These data suggest that PEX5-ATGL complex would translocate to the contact points between peroxisomes and LDs. Therefore, we tried to reflect these experimental data in the working model.

Minor points:

1. What is TMRE in New Supplementary Fig. 1g?

TMRE (tetramethylrhodamine ethyl ester) is a staining dye for mitochondria which is extensively used for labeling and measuring the membrane potential of mitochondria in living cells. We included this information in the legend of the new Supplementary Figure 1.

2. New Supplementary Fig. 2h: Is intensity of PMP70 statistically significant between cells treated with vehicle and WY-14643?

We included statistical analysis in the new Supplementary Figure 2g.

3. Two antibodies each for PMP70 and PEX5 are listed without describing any explanation on page 21, lines 469-471.

We included this information in the methods section (p. 21).

Reviewer #3 (Remarks to the Author):

*In this revised manuscript, Kong et al provides both new experiments and substantial textual revisions, which address all of my concerns in their first submission. One major concern from the first submission was whether peroxisomes themselves were required for fasting-induced lipolysis and not just PEX5. The authors have addressed this by showing that knockdown of *prx1*, *prx13* and *prx19* in *C.elegans* prevented lipid droplet reduction. This result suggests that the loss of peroxisomes reduces lipolysis during fasting in *C. elegans* as all three of these genes are peroxisome biogenesis factors. Therefore, the assumption here is that the loss of peroxisomes in mammalian cells may also result in reduction in lipolysis during fasting. However, in the revised manuscript, they suggest that the decrease in lipolysis and lack of peroxisome recruitment to lipid droplet in *PEX1* knockdown in adipose cells was due to the inability of peroxisomes to bind to the kinesin protein, *KifC*, which is involved in moving peroxisomes along the microtubule. There are several problems with this interpretation. First, while *PEX1* has been shown to bind to *KifC3* (Dietrich et al *BBA* (2013) 1833:2013), it was never shown to be an adaptor for *KifC3* in peroxisome mobility. Second, *PEX1* has been shown to be required in both *S. cerevisiae* and in cultured mammalian cells to prevent peroxisome degradation by autophagy (Nuttall et al *Autophagy* (2014) 10:835; Law et al *Autophagy* (2017) 13:868). Therefore, the lack of or decreased recruitment to lipid droplets may likely be due to the decreased number of*

peroxisomes. Although the authors show that PMP70 fluorescent intensity is not reduced in PEX1 knockdown in adipose cells, it is possible that there are decreased number of peroxisomes in these cells. This is because the knockdown of PEX1 does not prevent PMPs from targeting to peroxisomes and given that most PMPs have half-lives of less than 16 hours, it is not surprising that PMP70 fluorescents have not decreased within the duration of the RNAi knockdown assay. Furthermore, as pointed out in the previous review, fluorescent intensity is not a good measure of protein levels, especially when using confocal images. To demonstrate that PEX1 knockdown is NOT causing a decrease in peroxisome numbers, the authors should consider quantifying the number of peroxisomes in their adipose cells. Alternatively, the authors should consider removing their interpretation of the PEX1 knockdown data as a supporting evidence to the requirement of KifC3 for peroxisome recruitment to lipid droplet. It is the opinion of this reviewer that the KifB5 data is a sufficient negative control for the KifC3.

We really appreciate this comment and suggestion. To address this issue, we analyzed the number of peroxisomes in adipocytes. The quantification of peroxisome number was conducted as described by Law et al⁷. As shown in the Reviewer's only Figure 1a, suppression of PEX1 reduced the number of peroxisomes even though the number of peroxisomes was not largely altered by ISO treatment. Moreover, we noticed that the size of peroxisomes appeared to be increased by PEX1 suppression (Reviewer's only Figure 1b). As the reviewer pointed out, we cannot exclude the possibility that reduction in stimulated lipolysis and peroxisomal recruitment to LDs by PEX1 suppression might be due to the decreased number of peroxisomes. Thus, according to the reviewer's suggestion, we removed previous Supplementary Figure 2c and moved previous Supplementary Figure 2d and e to new Supplementary Figure 8a and b, respectively. Also, we modified the description of PEX1 knockdown data in the revised manuscript (pp. 6~7 and p. 17).

Reviewer's only Figure 1. The alteration of peroxisomal number and size by PEX1 suppression upon ISO.

a, Quantification of immunostained PMP70 positive peroxisome density in differentiated adipocytes. (number of PMP70 puncta per volume of each cell (number/μm³)). **b**, Quantification of PMP70 fluorescence positive peroxisomes that were classified by size using LAS X software. n=10-15 cells for each group. Cells were transfected with siNC or siPEX1. CON, control; ISO, isoproterenol. Cells were treated with ISO (1 μM) for 1 h. Data represent the mean ± SD; *P < 0.05 vs. siNC-CON in two-way ANOVA followed by Turkey's post-hoc test.

References

1. Zhang, J. *et al.* A tuberous sclerosis complex signalling node at the peroxisome regulates mTORC1 and autophagy in response to ROS. *Nat Cell Biol* **15**, 1186-1196 (2013).
2. Gatto, G.J., Jr., Geisbrecht, B.V., Gould, S.J. & Berg, J.M. Peroxisomal targeting signal-1 recognition by the TPR domains of human PEX5. *Nat Struct Biol* **7**, 1091-1095 (2000).
3. van der Klei, I.J. & Veenhuis, M. PTS1-independent sorting of peroxisomal matrix proteins by Pex5p. *Biochim Biophys Acta* **1763**, 1794-1800 (2006).
4. Klein, A.T., van den Berg, M., Bottger, G., Tabak, H.F. & Distel, B. *Saccharomyces cerevisiae* acyl-CoA oxidase follows a novel, non-PTS1, import pathway into peroxisomes that is dependent on Pex5p. *J Biol Chem* **277**, 25011-25019 (2002).
5. Elgersma, Y., van Roermund, C.W., Wanders, R.J. & Tabak, H.F. Peroxisomal and mitochondrial carnitine acetyltransferases of *Saccharomyces cerevisiae* are encoded by a single gene. *EMBO J* **14**, 3472-3479 (1995).
6. Gunkel, K., van Dijk, R., Veenhuis, M. & van der Klei, I.J. Routing of *Hansenula polymorpha* alcohol oxidase: an alternative peroxisomal protein-sorting machinery. *Mol Biol Cell* **15**, 1347-1355 (2004).
7. Law, K.B. *et al.* The peroxisomal AAA ATPase complex prevents pexophagy and development of peroxisome biogenesis disorders. *Autophagy* **13**, 868-884 (2017).

REVIEWERS' COMMENTS:

Reviewer #2 (Remarks to the Author):

Remarks to the Authors:

In this re-revised manuscript, the authors improved the manuscript by revising the text and figures. The reviewer #2 still has several concerns and comments to the Authors' replies as summarized below:

A) Comment to the authors' reply to Major point 1)

In response to the first half of this comment, the authors provided details of experimental condition and more explanation for the data. In a newly added text in Results section on page 11, lines 258-260, the reviewer #2 could not find the data showing the augmentation of lipolysis with FSK by overexpressing ATGL and PEX5 in Supplementary Fig. 5c. Additionally, the reviewer #2 has a concern why the authors used COS-7 cells for the analysis of glycerol release in Fig. 6e, despite the authors' careful examination for physical interaction and colocalization of ATGL with PEX5 assessed in HEK293 cells. This is now disclosed for the first time in this revised manuscript. For the latter part of Major point 1), the authors chose to exclude the data of "Reviewers only Figure 2" that would provide the readers valuable information on the interaction of ATGL with PEX5. The reviewer #2 disagrees with the reply that "We believe that the essential information of the direct interaction between ATGL and PEX5 has already been shown in the co-immunoprecipitation experiments (Figure 6)." Because co-immunoprecipitation experiments as well as proximity ligation assay (PLA) do not show "direct" interaction and cannot exclude the possibility of the indirect interaction mediated by additional protein(s), which is pointed out in the major comment No. 5 of the reviewer #2 in the 1st review and the major point No. 1 of the reviewer #2 in the 2nd review. Direct interaction should be evaluated by in vitro binding assay using recombinant proteins. The reviewer #2 thinks that the mode of interaction between ATGL and PEX5 is essential for establishing the authors' working model that PEX5 translocates ATGL upon fasting to the peroxisome-LD contact points. This assessment clearly becomes essential when "Reviewers only Figure 2" is not included in the main text.

B) Comment to the authors' reply to Major point 2)

Regarding to the role of phosphorylation of PEX5 in ATGL translocation, the authors adequately revised the working model of Figure 8 and its explanation.

Reviewer #3 (Remarks to the Author):

All my concerns have been addressed.

Response to the Reviewers' Comments

MS ID#: NCOMMS-18-6227380B

MS TITLE: Spatiotemporal contact between peroxisomes and lipid droplets regulates fasting-induced lipolysis via PEX5

Reviewers' comments:

Reviewer #2 (Remarks to the Author):

Remarks to the Authors:

In this re-revised manuscript, the authors improved the manuscript by revising the text and figures. The reviewer #2 still has several concerns and comments to the Authors' replies as summarized below:

A) Comment to the authors' reply to Major point 1)

In response to the first half of this comment, the authors provided details of experimental condition and more explanation for the data. In a newly added text in Results section on page 11, lines 258-260, the reviewer #2 could not find the data showing the augmentation of lipolysis with FSK by overexpressing ATGL and PEX5 in Supplementary Fig. 5c. Additionally, the reviewer #2 has a concern why the authors used COS-7 cells for the analysis of glycerol release in Fig. 6e, despite the authors' careful examination for physical interaction and colocalization of ATGL with PEX5 assessed in HEK293 cells. This is now disclosed for the first time in this revised manuscript.

For the latter part of Major point 1), the authors chose to exclude the data of "Reviewers only Figure 2" that would provide the readers valuable information on the interaction of ATGL with PEX5. The reviewer #2 disagrees with the reply that "We believe that the essential information of the direct interaction between ATGL and PEX5 has already been shown in the co-immunoprecipitation experiments (Figure 6)." Because co-immunoprecipitation experiments as well as proximity ligation assay (PLA) do not show "direct" interaction and cannot exclude the possibility of the indirect interaction mediated by additional protein(s), which is pointed out in the major comment No. 5 of the reviewer #2 in the 1st review and the major point No. 1 of the reviewer #2 in the 2nd review. Direct interaction should be evaluated by in vitro binding assay using recombinant proteins. The reviewer #2 thinks that the mode of interaction between ATGL and PEX5 is essential for establishing the authors' working model that PEX5 translocates ATGL upon fasting to the peroxisome-LD contact points. This assessment clearly becomes essential when "Reviewers only Figure 2" is not included in the main text.

Thank you for careful reading and constructive comment. During the revising process, the "Supplementary Figure 5c" was incorrectly inserted on page 12, lines 257-258. We removed this in the revised manuscript (p. 12).

The reviewer #2 asked the reason for using COS-7 cells in the analysis of glycerol release. Although it has not been properly reported, it seems that HEK293 cells would not be active for lipid metabolism. Instead, several key laboratories, including Drs. R. Zechner (University of Graz), H. Sul (UC Berkeley), and V. Puri (Ohio University), have utilized COS-7 cells to measure ATGL-mediated lipolytic activities¹⁻⁴. Thus, we have performed the glycerol release assay in COS-7 cells.

The point for the "direct interaction between ATGL and PEX5" is well taken. We also agree with the comment that co-IP and PLA assay cannot exclude the possibility of potential indirect interaction between PEX5 and ATGL. According to this comment, we modified the text to tone down some of our claims for the interaction between ATGL and PEX5 (p. 2, pp. 11~12, pp. 15~16, and p. 39).

Although the reviewer #2 asked to include the “Reviewer’s only Figure 2” from 1st reply letter in the main manuscript, we have a strong concern that including this figure would bring a wrong and confusing message to the peroxisome research community. As described in the answer for the 2nd reply letter comment No. 1 of the reviewer #2, it has been controversial whether an internal ARL sequence would act as a noncanonical PTS1 targeting signal. Also, to test and prove which sequence motif of ATGL would be responsible for interacting with PEX5 requires huge efforts, which would be an independent future study. Moreover, “Reviewer’s only Figure 2” does not support the argument for the “direct interaction” between ATGL and PEX5. Therefore, we are very cautious to move the previous Reviewer’s only Figure 2 into the main data.

B) Comment to the authors’ reply to Major point 2)

Regarding to the role of phosphorylation of PEX5 in ATGL translocation, the authors adequately revised the working model of Figure 8 and its explanation.

Reviewer #3 (Remarks to the Author):

All my concerns have been addressed.

References

1. Zimmermann, R. *et al.* Fat mobilization in adipose tissue is promoted by adipose triglyceride lipase. *Science* **306**, 1383-1386 (2004).
2. Villena, J.A., Roy, S., Sarkadi-Nagy, E., Kim, K.H. & Sul, H.S. Desnutrin, an adipocyte gene encoding a novel patatin domain-containing protein, is induced by fasting and glucocorticoids: ectopic expression of desnutrin increases triglyceride hydrolysis. *J Biol Chem* **279**, 47066-47075 (2004).
3. Grahn, T.H. *et al.* Fat-specific protein 27 (FSP27) interacts with adipose triglyceride lipase (ATGL) to regulate lipolysis and insulin sensitivity in human adipocytes. *J Biol Chem* **289**, 12029-12039 (2014).
4. Duncan, R.E., Ahmadian, M., Jaworski, K., Sarkadi-Nagy, E. & Sul, H.S. Regulation of lipolysis in adipocytes. *Annu Rev Nutr* **27**, 79-101 (2007).